# Study on the Tidal Dynamics of the Korea Strait Using the Extended Taylor Method

**Di Wu[1], Guohong Fang[1,2], Zexun Wei[1,2], Xinmei Cui[1,2]**

[1]First Institute of Oceanography, Ministry of Natural Resources, Qingdao, 266061, China

[2]Laboratory for Regional Oceanography and Numerical Modeling, Pilot National Laboratory for Marine Science and Technology, Qingdao, 266237, China

*Correspondence*: Guohong Fang (fanggh@fio.org.cn)

**Abstract**. The Korea Strait (KS) is a major navigation passage linking the Japan Sea (JS) to the East China Sea and Yellow Sea. Almost all existing studies on the tides in the KS employed either data analysis or numerical modelling methods; thus,

theoretical research is lacking. In this paper, we idealize the KS-JS basin as three connected uniform-depth rectangular areas and establish a theoretical model for the tides in the KS and JS using the extended Taylor method. The model-produced $K_1$ and $M_2$ tides are consistent with the satellite altimeter and tidal gauge observations, especially for the locations of the amphidromic points in the KS. The model solution provides the following insights into the tidal dynamics. The tidal system in each area can be decomposed into two oppositely travelling Kelvin waves and two families of Poincaré modes, with Kelvin waves

dominating the tidal system. The incident Kelvin wave can be reflected at the connecting cross-section, where abrupt increases in water depth and basin width occur from the KS to JS. At the connecting cross-section, the reflected wave has a phase-lag increase relative to the incident wave by less than 180°, causing the formation of amphidromic points in the KS. The above phase-lag increase depends on the angular frequency of the wave and becomes smaller as the angular frequency decreases. This dependence explains why the $K_1$ amphidromic point is located farther away from the connecting cross-section in

comparison to the $M_2$ amphidromic point.

## 1 Introduction

The Korea Strait (KS, also called the Tsushima Strait) connects the East China Sea (ECS) on southwest and the Japan Sea (the JS, also called the East Sea, or the Sea of Japan) on northeast. It is the main route linking the JS to the ECS and Yellow Sea and is thus an important passage for navigation. The strait is located on the continental shelf, and it has a length of

approximately 350 km, a width of 250 km, and an average water depth of approximately 100 m. The JS, which is adjacent to the KS, is a vast deep basin that has an average depth of approximately 2000 m and a depth of more than 3000 m at its deepest part. A sharp continental slope separates the KS and the JS, and it presents abrupt depth and width changes (Fig. 1). Such topographic characteristics create the unique tidal waves that occur in the KS.



Ogura (1933) first conducted a comprehensive study of the tides in the seas adjacent to Japan using data from the tidal stations along the coast and gained a preliminary understanding of the characteristics of the tides, including amphidromic systems in the KS. Since then, many researchers have investigated the tides in the strait via observations (Odamaki, 1989a; Matsumoto et al., 2000; Morimoto et al., 2000; Teague et al., 2001; Takikawa et al., 2003) and numerical simulations (Fang and Yang, 1988; Kang et al., 1991; Choi et al., 1999; Book et al., 2004). The results of these studies show consistent structures of the tidal waves in the KS. Fig. 2 displays the distributions of the $K_1$ and $M_2$ tidal constituents based on the global tidal model DTU10, which is based on satellite altimeter observations (Cheng and Andersen, 2011). The figures show that the amplitudes of the diurnal tides are smaller than the semidiurnal tides. The peak amplitude of the semidiurnal tide appears on the south coast of South Korea, and lower amplitudes occur along the southern shore of the strait from the ECS to the JS. Distinguishing features include (1) $K_1$ and $M_2$ amphidromic points in the strait that appear in the northeast part of the KS close to the southern coast of the Korean Peninsula; and (2) the $M_2$ amphidromic point appears further northeast and closer to the shelf break relative to the $K_1$ tide.

However, almost all previous studies have employed either data analysis or numerical modelling methods; thus, theoretical research is lacking. In particular, the existence of amphidromic points in the northeast KS for both diurnal and semidiurnal tides has not been explained based on geophysical dynamics. In this paper, we intend to establish a theoretical model for the $K_1$ and $M_2$ tides in the KS-JS basin using the extended Taylor method. The model idealizes the KS-JS basin into three connected uniform-depth rectangular areas, with the effects of bottom friction and Coriolis force included in the governing equations and with the observed tides specified as open boundary conditions. The extended Taylor method enables us to obtain analytical solutions consisting of a series of Kelvin waves and Poincaré modes.



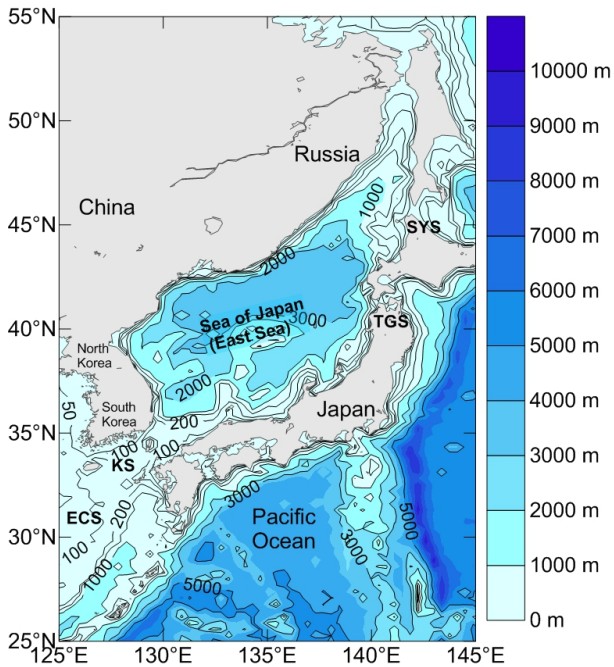

**Figure 1: Map of the Korea Strait and its neighbouring areas. (SYS- Soya Strait, TGS- Tsugaru Strait, KS- Korea Strait, ECS-East China Sea). Isobaths are in metres (based on ETOPO1 from US National Geophysical Center).**

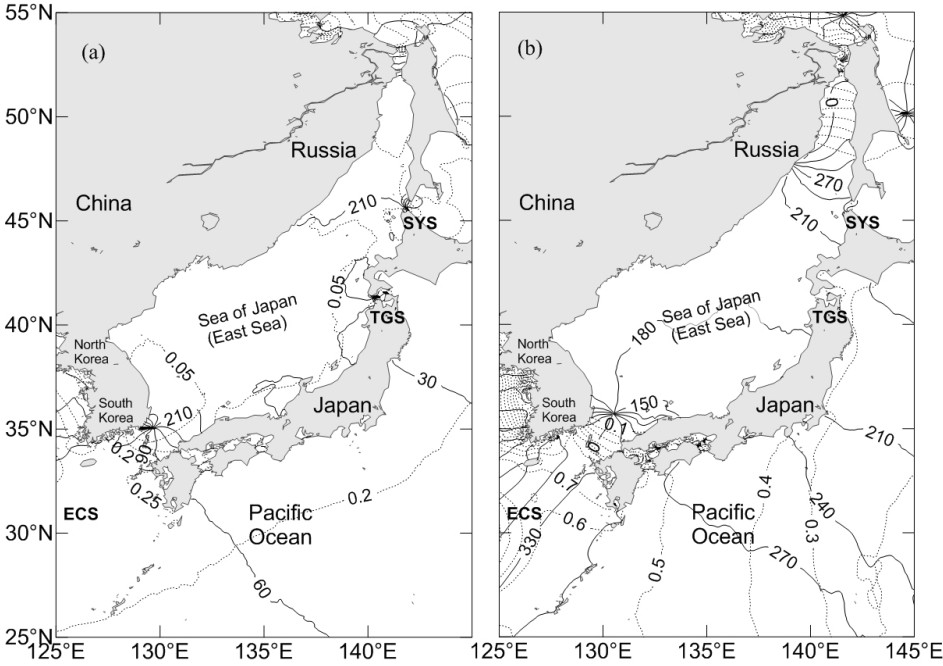

5    **Figure 2: Tidal charts of the KS and its neighbouring areas based on DTU10 (Cheng and Andersen, 2011) for the (a) K1 tide and (b) M2 tide. Solid lines represent the Greenwich phase lag (in degrees), and dashed lines represent amplitude (in metres).**



## 2 The extended Taylor method and its application to multiple rectangular areas

The Taylor problem is a classic tidal dynamic problem (Hendershott and Speranza, 1971). Taylor (1922) first presented an analytical solution for tides in a semi-infinite rotating rectangular channel of uniform depth to explain the formation of amphidromic systems in gulfs and applied the theory to the North Sea. The classic Taylor problem was subsequently improved

by introducing frictional effects (Fang and Wang, 1966; Webb, 1976; Rienecker and Teubner, 1980) and open boundary conditions (Fang et al., 1991) to study tides in multiple rectangular basins (Jung et al., 2005; Roos and Schuttelaars, 2011; Roos et al., 2011) as well as to solve tidal dynamics in a strait (Wu et al., 2018).

The method initiated by Taylor and developed afterwards is called the extended Taylor method (Wu et al., 2018). This method is especially useful in understanding the tidal dynamics in marginal seas and straits because the tidal waves in these

sea areas can generally be represented by combinations of the Kelvin waves and Poincaré waves/modes (e. g., Taylor, 1922; Fang and Wang, 1966; Hendershott and Speranza, 1971; Webb, 1976; Fang et al., 1991; Carbajal, 1997; Jung et al., 2005; Roos and Schuttelaars, 2011; Roos et al., 2011; Wu et al., 2018).

### 2.1 Governing equations and boundary conditions for multiple rectangular areas

A sketch of the model geometry is shown in Fig. 3, and it consists of a sequence of $J$ rectangular areas with length $L_j$, width

$W_j$ and uniform depth $h_j$ for the $j$th rectangular area (denoted as Area$j$, $j=1, …, J$). For convenience, the shape of the study region shown in Fig. 3 is the same as that for the idealized KS–JS basin, which will be described in the next section. In particular, Area1 represents the KS, which is our focus area in this study.

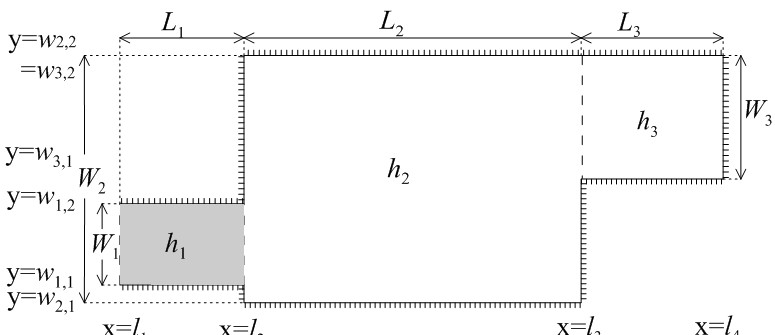

**Figure 3: Model geometry.**

Consider a tidal wave of angular frequency $\sigma$ and typical elevation amplitude $H$. We assume $H/h \ll 1$, and the conservation of momentum and mass leads to the following depth-averaged linear shallow water equations on the $f$ plane:




$$\begin{cases} \frac{\partial \tilde{u}_j}{\partial t} - f_j \tilde{v}_j = -g \frac{\partial \tilde{\zeta}_j}{\partial x} - \gamma_j \tilde{u}_j \\ \frac{\partial \tilde{v}_j}{\partial t} + f_j \tilde{u}_j = -g \frac{\partial \tilde{\zeta}_j}{\partial y} - \gamma_j \tilde{v}_j, \\ \frac{\partial \tilde{\zeta}_j}{\partial t} = -h_j \left[ \frac{\partial \tilde{u}_j}{\partial x} + \frac{\partial \tilde{v}_j}{\partial y} \right] \end{cases} \tag{1}$$

where $x$ and $y$ are coordinates in the longitudinal (along-channel) and transverse (cross-channel) directions; $t$ represents time; $\tilde{u}_j$ and $\tilde{v}_j$ represent the depth-averaged flow velocity components in the $x$ and $y$ directions, respectively, with the subscript $j$ indicating the area number; $\tilde{\zeta}_j$ represents the free surface elevation above the mean level; $\gamma_j$ represents the

frictional coefficient, which is taken as a constant for each tidal constituent in each area; $g =9.8 \text{ ms}^{-2}$ represents the acceleration due to gravity; and $f_j$ represents the Coriolis parameter, which is also taken as a constant based on the average of the concerned area. The equations in (1) for each $j$ are two-dimensional linearized shallow water equations on an $f$-plane with the momentum advection neglected. For any $j$, the equations are the same as those used in the work of Taylor (1922) except that bottom friction is now incorporated, such as in Fang and Wang (1966), Webb (1976), Rienecker and Teubner (1980),

etc. When a monochromatic wave is considered, $\left( \tilde{\zeta}_j, \tilde{u}_j, \tilde{v}_j \right)$ can be expressed as follows:

$$\left( \tilde{\zeta}_j, \tilde{u}_j, \tilde{v}_j \right) = \text{Re}\left( \zeta_j, u_j, v_j \right) e^{i\sigma t}, \tag{2}$$

where Re stands for the real part of the complex quantity that follows, $\left( \zeta_j, u_j, v_j \right)$ are referred to as complex amplitudes of $\left( \tilde{\zeta}_j, \tilde{u}_j, \tilde{v}_j \right)$, respectively, $i \equiv \sqrt{-1}$ is the imaginary unit, and $\sigma$ is the angular frequency of the wave. For this wave, Eq. (1) can be reduced as follows:

$$\begin{cases} \left( \mu_j + i \right) u_j - v_j v_j = -\frac{g}{\sigma} \frac{\partial \zeta_j}{\partial x} \\ \left( \mu_j + i \right) v_j + v_j u_j = -\frac{g}{\sigma} \frac{\partial \zeta_j}{\partial y}, \\ \zeta_j = \frac{i h_j}{\sigma} \left[ \frac{\partial u_j}{\partial x} + \frac{\partial v_j}{\partial y} \right] \end{cases} \tag{3}$$

in which

$$\mu_j = \frac{\gamma_j}{\sigma} \quad \text{and} \quad v_j = \frac{f_j}{\sigma} \ . \tag{4}$$

Provided that the $j$-th rectangular area, denoted as Area$j$, has a width of $W_j$, a length of $L_j$, and ranges from $x = l_j$ to $x = l_{j+1}$ $(l_{j+1} = l_j + L_j)$ in the $x$ direction and from $y = w_{j,1}$ to $y = w_{j,2}$ $(w_{j,2} = w_{j,1} + W_j)$ in the $y$ direction, the boundary

conditions along the sidewalls within $x \in [l_j, l_{j+1}]$ are taken as follows:

$$v_j = 0 \ \text{at} \ y = w_{j,1} \ \text{and} \ y = w_{j,2}. \tag{5}$$

Along the cross-sections, such as $x = l_j$, various choices of boundary conditions are applicable depending on the problem:

$$u_j = 0, \tag{6}$$

if the cross-section is a closed boundary;

$$u_j = \pm \sqrt{\frac{g}{(1-i\mu_j)h_j}} \zeta_j, \tag{7}$$

if the free radiation in the positive/negative $x$ direction occurs on the cross-section;

$$\zeta_j = \hat{\zeta}_j, \tag{8}$$

if the tidal elevation is specified as $\hat{\zeta}_j$ along the cross-section; and

$$\begin{cases} \zeta_j = \zeta_{j+1}, \\ u_j h_j = u_{j+1} h_{j+1}, \end{cases} \tag{9}$$

if the cross-section is a connecting boundary of the areas $j$ and $j + 1$, with each having a different uniform depth of $h_j$ and $h_{j+1}$.

Equation (9) is matching conditions accounting for sea level continuity and volume transport continuity. The individual Eqs.





(6) to (9), or their combination, may be used as boundary conditions for the cross-sections. The relationship between $u_j$ and $\zeta_j$ shown in Eq. (7) is based on the solution for progressive Kelvin waves in the presence of friction, which will be given in Eqs. (10) and (11) below.

**2.2 General solution**

For the $j$-th rectangular area, that is, for $x \in [l_j, l_{j+1}]$ and $y \in [w_{j,1}, w_{j,2}]$, the governing equations in Eq. (3) only have the following four forms satisfying the sidewall boundary condition of Eq. (5) (see, e. g., Fang et al. 1991):

$$\begin{cases} v_{j,1} = 0, \\ u_{j,1} = -a_j \exp[\alpha_j y + i\beta_j(x - l_j)], \\ \zeta_{j,1} = \frac{\beta_j}{\sigma} h_j a_j \exp[\alpha_j y + i\beta_j(x - l_j)]; \end{cases} \tag{10}$$

$$\begin{cases} v_{j,2} = 0, \\ u_{j,2} = b_j \exp[-\alpha_j y - i\beta_j(x - l_j)], \\ \zeta_{j,2} = \frac{\beta_j}{\sigma} h b_j \exp[-\alpha_j y - i\beta_j(x - l_j)]; \end{cases} \tag{11}$$

$$\begin{cases} v_{j,3} = \sum_{n=1}^{\infty} \kappa_{j,n} \sin r_{j,n} y \exp[-s_{j,n}(x - l_j)], \\ u_{j,3} = \sum_{n=1}^{\infty} \kappa_{j,n}(A_{j,n} \cos r_{j,n} y + B_{j,n} \sin r_{j,n} y) \exp[-s_{j,n}(x - l_j)], \\ \zeta_{j,3} = \frac{ih_j}{\sigma} \sum_{n=1}^{\infty} \kappa_{j,n}(C_{j,n} \cos r_{j,n} y + D_{1,n} \sin r_{j,n} y) \exp[-s_{j,n}(x - l_j)]; \end{cases} \tag{12}$$

and

$$\begin{cases} v_{j,4} = \sum_{n=1}^{\infty} \lambda_{j,n} \sin r_{j,n} y \exp[-s_{j,n}(l_{j+1} - x)], \\ u_{j,4} = \sum_{n=1}^{\infty} \lambda_{j,n}(A'_{j,n} \cos r_{j,n} y + B'_{j,n} \sin r_{j,n} y) \exp[-s_{j,n}(l_{j+1} - x)], \\ \zeta_{j,4} = \frac{ih_j}{\sigma} \sum_{n=1}^{\infty} \lambda_{j,n}(C'_{j,n} \cos r_{j,n} y + D'_{j,n} \sin r_{j,n} y) \exp[-s_{j,n}(l_{j+1} - x)]. \end{cases} \tag{13}$$

where $\alpha_j$, $\beta_j$, $r_{j,n}$ and $s_{j,n}$ are equal to the following:

$$\alpha_j = \frac{v_j}{(1 - i\mu_j)^{1/2}} k_j, \tag{14}$$

$$\beta_j = (1 - i\mu_j)^{1/2} k_j, \tag{15}$$

$$r_{j,n} = \frac{n\pi}{W_j}, \tag{16}$$

and

$$s_{j,n} = (r_{j,n}^2 + \alpha_j^2 - \beta_j^2)^{\frac{1}{2}}, \tag{17}$$

in which $k_j = \sigma/c_j$ is the wave number, with $c_j = \sqrt{gh_j}$ being the wave speed of the Kelvin wave in the absence of friction. The parameters $s_{j,n}$ in Eq. (17) are of fundamental importance in determining the characteristic of the Poincaré modes. If

$\mathrm{Re}(\beta_j^2 - \alpha_j^2)^{1/2} < \pi/W_j$, all Poincaré modes are bound in the vicinity of the open, connecting or closed cross-sections (see Fang and Wang, 1966; Hendershott and Speranza, 1971 for in absence of friction); while if $\mathrm{Re}(\beta_j^2 - \alpha_j^2)^{1/2} > n\pi/W_j$, the $n$-th and lower-order Poincaré modes are propagating waves. In the present study, the inequality $\mathrm{Re}(\beta_j^2 - \alpha_j^2)^{1/2} < \pi/W_j$ holds for both the idealized KS and JS, so that all Poincaré modes in the present study appear in a bound form. The parameter $s_{j,n}$




has two complex values for each $n$, and here,we choose the one that has a positive real part. To satisfy the equations in Eq.

(3), $(A_{j,n}, B_{j,n}, C_{j,n}, D_{j,n})$ and $(A'_{j,n}, B'_{j,n}, C'_{j,n}, D'_{j,n})$ should be as follows:

$$A_{j,n} = \frac{\left[(\mu_j+i)^2+\nu_j^2\right]r_{j,n}s_{j,n}}{(\mu_j+i)^2 r_{j,n}^2+\nu_j^2 s_{j,n}^2} s_{j,n}, \tag{18}$$

$$B_{j,n} = \frac{\nu_j(\mu_j+i)\left(\alpha_j^2-\beta_j^2\right)}{(\mu_j+i)^2 r_{j,n}^2+\nu_j^2 s_{j,n}^2}, \tag{19}$$

$$C_{j,n} = r_{j,n} - s_{j,n}A_{j,n}, \tag{20}$$

$$D_{j,n} = -s_{j,n}B_{j,n}, \tag{21}$$

$$A'_{j,n} = -A_{j,n}, \tag{22}$$

$$B'_{j,n} = B_{j,n}, \tag{23}$$

$$C'_{j,n} = C_{j,n}, \tag{24}$$

and

$$D'_{j,n} = -D_{j,n} \tag{25}$$

Equations (10) and (11) represent Kelvin waves propagating in the $-x$ and $x$ directions, respectively; and Eqs. (12) and (13) represent two families of Poincaré modes bound along the cross-sections $x = l_j$ and $l_{j+1}$ in the $j$-th rectangular area, respectively. Coefficients $(a_j, b_j, \kappa_{j,n}, \lambda_{j,n})$ determine amplitudes and phase lags of Kelvin waves and Poincaré modes. These

coefficients must be chosen to satisfy the boundary conditions, using preferably the collocation approach.

### 2.3 Defant's collocation approach

The collocation approach was first proposed by Defant in 1925 (see Defant, 1961), and is convenient in determining the coefficients $(a_j, b_j, \kappa_{j,n}, \lambda_{j,n})$. In the simplest case, that is, if the model domain contains only a single rectangular area, then $J = 1$ and the index $j$ has only one value: $j = 1$, the calculation procedure can be as follows. First, we truncate each of the two

families of Poincaré modes in Eqs. (12) and (13) at the $N_1$-th order so that the number of undetermined coefficients for two families of Poincaré modes is $2N_1$ and the total number of undetermined coefficients (plus those for a pair of Kelvin waves) is thus $2N_1 + 2$. To determine these unknowns, we take equally spaced $N_1 + 1$ dots, which are called collocation points, located at $y = w_{1,1} + \frac{W_1}{2(N_1+1)}, \ w_{1,1} + \frac{3W_1}{2(N_1+1)}, \ ... \ , \ w_{1,1} + \frac{(2N_1+1)W_1}{2(N_1+1)}$ on both cross-sections $x = l_1$ and $l_2$. At these points, one of the boundary conditions given by Eqs. (6) to (8) should be satisfied, which yields $2N_1 + 2$ equations. By solving this

system of equations, we can obtain $2N_1 + 2$ coefficients $(a_1, b_1, \kappa_{1,n}, \lambda_{1,n})$. Because the high-order Poincaré modes, which have great values of $n$ and $s_{1,n}$ in Eqs. (12) and (13), decay from the boundary very quickly, it is generally necessary to retain only a few lower-order terms. In the above single-rectangle case, the spacing of collocation points is equal to $\Delta y = W_1/(N_1 + 1)$.

For $J > 1$, that is, the model contains multiple rectangular areas connected one by one, we can treat the approach in the





following way. First, we may choose a common divisor of $W_1, W_2, \dots, W_J$ as a common spacing, which is denoted by $\Delta y$, for all areas. For the $j$th rectangle (Fig. 3), we may select the collocation points at $y = w_{j,1} + \frac{\Delta y}{2}$, $w_{j,1} + \frac{3\Delta y}{2}$, $\dots$, $w_{j,2} - \frac{\Delta y}{2}$ on the cross-sections $x = l_j$ and $x = l_{j+1}$, where $w_{j,2} = w_{j,1} + W_j$. The number of collocation points on each cross-section in this area is equal to $W_j / \Delta y$. Thus the number of undetermined coefficients for the Poincaré modes is selected to be

$N_j = (W_j / \Delta y) - 1$. Accordingly, there will be in total $\sum_{j=1}^{J}(2N_j + 2)$ collocation points in $J$ areas. Note that on the cross-section connecting Area$j$ and Area($j$+1), the collocation points that belong to Area$j$ and those that belong to Area($j$+1) are located at the same positions. For the points located on the open or closed boundaries, Eqs. (6) to (8) are applicable, while for the points located on the cross-sections connecting two areas, Eq. (9) should be applied. From these $\sum_{j=1}^{J}(2N_j + 2)$ equations, we can obtain $\sum_{j=1}^{J}(2N_j + 2)$ coefficients $(a_j, b_j, \kappa_{j,n}, \lambda_{j,n})$, in which $j = 1, 2, \dots, J$ and $n = 1, 2, \dots N_j$.

**3 Tidal dynamics of the Korea Strait**

As noted by Odamaki (1989b), the co-oscillating tides are dominant in the JS, which is mainly induced by inputs at the opening of the KS rather than those through the TGS and SYS. Furthermore, our study focuses on the KS, in which influences of the tide-generating force and the inputs from the TGS and SYS are negligible. Therefore, we idealize the KS-JS basin as a semi-enclosed basin with a sole opening connected to the ECS and study the co-oscillating tides generated by the tidal waves from

the ECS through the opening.

**3.1 Model configuration and parameters for the Korea Strait and Japan Sea**

To establish an idealized analytical model for the KS–JS basin, we use three rectangular areas as shown in Fig. 4 to represent the study region. The first rectangle, designated as Area1, represents the KS, which is our area of focus. According to the shape of its coastline, we use two rectangles designated as Area2 and Area3 to represent the JS. We place the x-axis parallel to but

200 km away from the southeast sidewall of the KS (that is, $w_{1,1}$ in Fig. 3 is equal to 200 km), and the y-axis is in the direction perpendicular to the x-axis through the opening of the KS (Fig. 4). The selected depths are the mean depths calculated based on the topographic dataset ETOPO1. The $K_1$ and $M_2$ angular frequencies are equal to $7.2867 \times 10^{-5} \text{s}^{-1}$ and $1.4052 \times 10^{-4} \text{s}^{-1}$, respectively. The details of the model parameters can be found in Table 1.





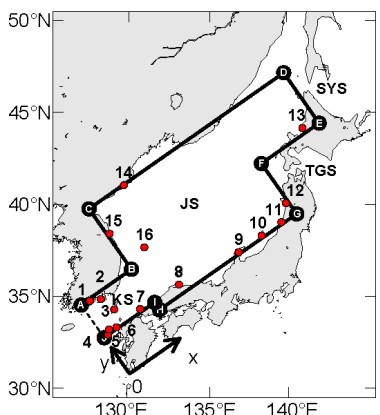

**Figure 4: Idealized model domain fitting the Korea Strait and Japan Sea. The dashed line represents open boundary, and the solid lines represent closed boundaries. A, B, … , J indicate the localities of the points used in Fig. 6 for model-observation comparison. Numbered red dots are tidal gauge stations where the observed harmonic constants are used for model validation in Table 2.**

Based on the depths listed in Table 1, the wavelengths of the $K_1$ Kelvin waves in these three areas are 2686 km, 12189 km, and 11398 km, respectively, and those of the $M_2$ Kelvin waves are 1393 km, 6321 km, and 5911 km, respectively. Because the widths of the areas are all smaller than half the corresponding Kelvin wavelengths, the inequality $\mathrm{Re}(\beta_j^2 - \alpha_j^2) < \pi/W_j$ as stated in the subsection 2.2 is satisfied (see also Godin, 1965; Fang and Wang, 1966; Wu et al., 2018), Thus the Poincaré modes can only exist in a bound form.

**Table 1**. Parameters used in the model.

| Parameter | Area1 | Area2 | Area3 |
|---|---|---|---|
| $W_j$ (km) | 230 | 700 | 350 |
| $L_j$ (km) | 350 | 950 | 400 |
| $w_{j,1}$ (km) | 250 | 200 | 550 |
| $f_j$ $(10^{-5}s^{-1})$ | 8.28 | 9.24 | 10.10 |
| $h_j$ (m) | 99 | 2039 | 1783 |
| $N_j$ | 22 | 69 | 34 |

In addition to the parameters listed in Table 1, we need to estimate the parameters $\mu_{M_2}$ and $\mu_{K_1}$ as defined by Eq. (4). Since $M_2$ has the largest tidal current in the KS (Teague et al., 2001), and we assume that the tidal currents are rectilinear, the linearized frictional coefficient for $M_2$ is approximately equal to the following, after Pingree and Griffiths (1981), Fang (1987) and Inoue and Garrett (2007),





$$\gamma_{M_2} \approx \frac{C_D}{h} \frac{8}{3\pi} U_{M_2} \left( 1 + \frac{3}{4} \sum_{i=2,3,\dots} \epsilon_i^2 \right) , \tag{26}$$

where $C_D$ is the drag coefficient and $U_{M_2}$ is the tidal current amplitude of M$_2$, $\epsilon_i = U_i / U_{M_2}$, with $U_i$ representing the tidal current amplitude of the constituent $i$ (here, we designate $i$=1 for M$_2$ and $i$=2, 3, … for any constituents other than M$_2$). According to Fang (1987) and Inoue and Garrett (2007), the linearized frictional coefficient for the non-dominant constituent

$i$ is approximately equal to the following:

$$\gamma_i \approx \frac{C_D}{h} \frac{4}{\pi} U_{M_2} \left( 1 + \frac{\epsilon_i^2}{8} + \frac{1}{4} \sum_{\substack{k=2,3,\dots \\ k \neq i}} \epsilon_k^2 \right) , \tag{27}$$

Inserting Eqs. (26) and (27) into Eq. (4), we can obtain the parameter $\mu$. Teague et al. (2001) provided tidal current harmonic constants at 10 mooring stations along two cross-sections in the KS. The averaged values of the major semi-axes of the tidal current ellipses at these stations are 0.154, 0.119, 0.101 and 0.062 m/s for M$_2$, K$_1$, O$_1$ and S$_2$, respectively. Here, we use these

values and $C_D \approx 0.0026$ to estimate the parameters in Eqs. (26) and (27). Then, after inserting these values into Eq. (4), we obtain rough estimates of $\mu_{M_2}$ and $\mu_{K_1}$ for the KS (Area1), which are approximately 0.05 and 0.09, respectively. Since the JS is much deeper and has much weaker tidal currents than the KS, we simply let $\mu_{K_1} = \mu_{M_2} = 0$ for both Area2 and Area3.

For the collocation approach, we take 10 km as the spacing between collocation points. Thus in this model, a total of 198 collocation points are used to establish 256 equations, and the parameters of 3 pairs of Kelvin waves and 125 pairs of Poincaré

modes can be obtained. Along the open boundary of the KS, the open boundary condition Eq. (8) is employed, with the value of $\hat{\zeta}$ equal to the observed harmonic constants from the global tide model DTU10 (Cheng and Anderson, 2011). Along the cross-sections connecting Area1 with Area2 and Area2 with Area3, the matching conditions Eq. (9) are applied. Along the solid cross-sections, condition Eq. (6) is used.

**3.2 Model results and validation**

The obtained analytical solutions of the K$_1$ and M$_2$ tides using the extended Taylor method are shown in Fig.5a and 5b, respectively. The maximum amplitude of the K$_1$ tide is 0.34 m, which appears at the southwest corner of the KS. The amplitude decreases from southwest to northeast, and a counter-clockwise tidal wave system occurs in the northeast part of the KS, with amplitudes less than 0.05 m near the amphidromic point. A co-tidal line with a phase lag of 210° runs from the amphidromic point in the KS into the southwest JS.





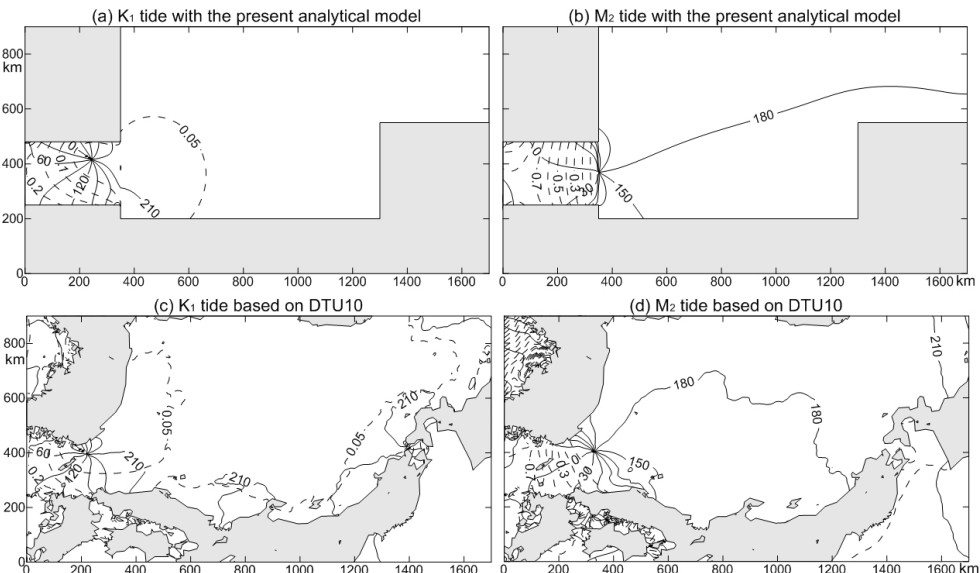

Figure 5: Comparison of tidal system charts. (a) $K_1$ and (b) $M_2$ tides from the present analytical model; and (c) $K_1$ and (d) $M_2$ tides from DTU10 (Chen and Andersen, 2011).

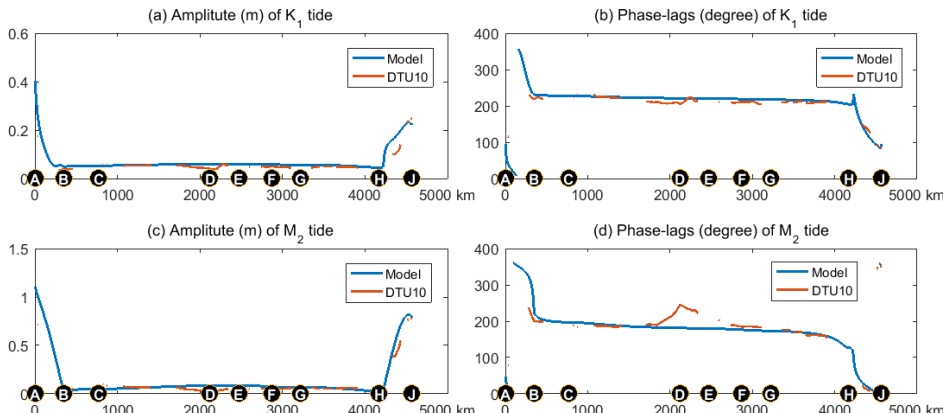

5 Figure 6: Comparison of model results (blue) and observations based on DTU10 (orange) along the coasts. (a) $K_1$ amplitudes; (b) $K_1$ phase lags; (c) $M_2$ amplitudes; and (d) $M_2$ phase lags. The locations of the points A, B, … , J are shown in Fig. 4.

The maximum amplitude of the $M_2$ tide is 1.02 m, which appears at the westernmost corner of the KS. The amplitude

decreases gradually from southwest to northeast along the direction of the strait, and the amphidromic point occurs at the

10 junction of the KS and JS. The amplitudes near the amphidromic point are lower than 0.1 m, and the phase lags in the most

part of the JS vary from 150° to 210°. The comparison with the tidal charts based on data from DTU10 (Fig.5c, d) shows that

the model-produced tidal systems agree fairly well with the observations.





To quantitatively validate the model results, we first extract the data along the solid boundary of the model for comparison as shown in Fig.6. For the $K_1$ tide, the model-produced amplitudes and phase lags along the boundary in the JS both agree well with the observed data, although small differences occur at the northern corner of the JS. For the $M_2$ tide, the greatest phase-lag errors are approximately 70° at the northernmost corner of the JS due to the existence of a degenerated amphidromic point near this area (Fig. 2b).

For further validation, we select 16 tide gauge stations where harmonic constants are available from the International Hydrographic Bureau (1930). The station locations are shown in Fig. 4. The result of the comparison is given in Table 2, which also shows that the model results are consistent with the data obtained from gauge observations: the RMS (root mean square) differences of amplitudes of $K_1$ and $M_2$ are 0.014 and 0.031 m, respectively; and those of the phase lags are 7.4° and 6.4°, respectively.

**Table 2**. Comparison between harmonic constants from the observations and models at coastal tide gauge stations.

| No | Station Name | $K_1$ | | | | $M_2$ | | | |
|----|--------------|-------|-------|-------|-------|-------|-------|-------|-------|
| | | Amplitude (m) | | Phase lag (°) | | Amplitude (m) | | Phase lag (°) | |
| | | obs | model | obs | model | obs | model | obs | model |
| 1 | Reisui Ko | 0.21 | 0.20 | 50 | 38 | 1.02 | 0.93 | 357 | 10 |
| 2 | Toei Ko | 0.16 | 0.11 | 46 | 38 | 0.80 | 0.77 | 355 | 2 |
| 3 | Takesiki Ko, Aso Wan | 0.12 | 0.11 | 83 | 87 | 0.66 | 0.66 | 1 | 6 |
| 4 | Aokata | 0.23 | 0.23 | 90 | 85 | 0.80 | 0.81 | 356 | 358 |
| 5 | Konoura, Uku Sima | 0.20 | 0.22 | 92 | 88 | 0.78 | 0.79 | 354 | 2 |
| 6 | Usuka Wan, Hirado Sima | 0.19 | 0.21 | 102 | 97 | 0.74 | 0.78 | 2 | 7 |
| 7 | Kottoi | 0.12 | 0.13 | 174 | 157 | 0.32 | 0.33 | 31 | 32 |
| 8 | Sitirui | 0.04 | 0.05 | 206 | 213 | 0.06 | 0.03 | 152 | 156 |
| 9 | Nakai Iri,Hoku Wan | 0.06 | 0.05 | 215 | 216 | 0.07 | 0.06 | 172 | 169 |
| 10 | Ryotu Ko, Sado | 0.05 | 0.06 | 211 | 217 | 0.05 | 0.06 | 181 | 172 |
| 11 | Kamo Ko | 0.06 | 0.06 | 211 | 217 | 0.07 | 0.07 | 174 | 173 |
| 12 | Akita | 0.06 | 0.06 | 220 | 217 | 0.05 | 0.07 | 174 | 174 |
| 13 | Hamamasu | 0.05 | 0.06 | 211 | 220 | 0.05 | 0.08 | 185 | 179 |
| 14 | Zyosin Ko | 0.06 | 0.05 | 227 | 226 | 0.08 | 0.05 | 187 | 194 |
| 15 | Sokcho | 0.04 | 0.05 | 236 | 228 | 0.07 | 0.05 | 189 | 199 |
| 16 | Uturyo To | 0.04 | 0.05 | 222 | 226 | 0.04 | 0.04 | 194 | 193 |
| | RMS difference | 0.014 | | 7.4 | | 0.031 | | 6.4 | |

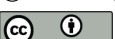
Although the theoretical model greatly simplifies the topography and boundary, the amplitude and phase-lag differences of these two tidal constituents are very small in the KS and its surroundings and the basic characteristics of the tidal patterns are well retained (Fig. 5). These findings show that the simplification of the model is reasonable and the extended Taylor method is appropriate for modelling the tides in the KS-JS basin. Therefore, it is meaningful to use the model results for theoretical

analysis.

### 3.3 Tidal waves in the Korea Strait

To reveal the relative importance of the Kelvin waves versus Poincaré modes in the modelled Korea Strait, the superposition of Kelvin waves and that of the Poincaré modes are given in the upper panels of Fig.7 for $K_1$ and in the upper panels of Fig.8 for $M_2$.

For the $K_1$ tide in the KS, the superposition of the incident (northeastward) and the reflected (southwestward) Kelvin waves appears as a counter-clockwise amphidromic system, with the amphidromic point located near the middle of the strait, but closer to the southeast coast of Korea (Fig.7a). The highest amplitude of the superposed Kelvin waves is 0.3 m, and the mean difference from the observations is less than 0.03 m. The superposition of all Poincaré modes has amplitudes of approximately 0.1 m near the cross-sections on both left and right sides, and a counter-clockwise amphidromic point exists nearly at the centre

of the strait (Fig. 7b). Since the amplitudes of the superposed Poincaré modes are significantly smaller than those of the superposed Kelvin waves, the latter can basically represent the total tidal pattern, including the counter-clockwise amphidromic system.

For the $M_2$ tide, the highest amplitude of the superposition of two Kelvin waves is approximately 0.96 m, which appears at the southwest corner of the strait (Fig. 8a). The amplitude decreases from southwest to northeast along the strait, and the

amphidromic point appears near the cross-section connecting to the JS, where a topographic step exists. The maximum deviation of the amplitudes of the superposed Kelvin waves from the observations is 0.06 m, and the structure of the superposed Kelvin waves is consistent with the observation. The amplitudes of the superposed Poincaré modes are generally less than 0.2 m on both left and right sides of the KS, and they decay rapidly towards the middle of the strait, thus forming a counter-clockwise amphidromic system structure (Fig. 8b). Therefore, the $M_2$ tide in the KS is also mainly controlled by Kelvin waves.

The above results show that the Poincaré modes only exist along the open boundary and the connecting cross-section and their amplitudes quickly approach to zero away from these cross-sections. In fact, these properties of the Poincaré wave are inherent in any narrow strait. Therefore, in the following, we will focus on Kelvin waves and analyze the characteristics of the incident (northeastward) and reflected (southwestward) Kelvin waves.

The incident and reflected $K_1$ Kelvin waves are shown in Figs. 7c and 7d, respectively. The area-mean amplitude of the

incident Kelvin wave in the KS is 0.253 m, and that of the reflected Kelvin wave is 0.196 m, which is 77% of the incident



Kelvin wave. On the connecting cross-section, the section-mean amplitude of the incident Kelvin wave is 0.249 m, and the section-mean phase lag is 151.3°. The section-mean amplitude of the reflected Kelvin wave is 0.199 m, which is 80% of the incident Kelvin wave. The section-mean phase lag is 294.2°, indicating that the phase lag increases by 142.9° when the wave is reflected. The amphidromic point of the superposed Kelvin wave is 145 km away from the step and close to the northwest

shore of the KS.

The incident and reflected $M_2$ Kelvin waves are shown in Figs. 8c and 8d, respectively. The area-mean amplitude of the incident Kelvin wave in the KS is 0.466 m, and that of the reflected Kelvin wave is 0.443 m, which is 95% of the incident Kelvin wave. This ratio is larger than the $K_1$ tide because the bottom friction of $M_2$ is smaller and less energy is lost in the propagation process. On the connecting cross-section, the mean amplitude of the incident Kelvin wave is 0.457 cm, and the

phase lag is 97.7°. The mean amplitude of the reflected Kelvin wave is 0.452 m, which is 99% of the incident Kelvin wave, and the phase lag is approximately 266.8°, with a phase-lag increase of 169.1°, which is closer to 180° as compared to the corresponding value of the $K_1$ tide. Accordingly, the $M_2$ amphidromic point of the superposed Kelvin wave shifts to approximately 20 km away from the step. A comparison between Fig. 7a and Fig. 8a shows that the amphidromic point of $K_1$ is located west of that of $M_2$. This result reproduces well the observed phenomenon as seen from Fig. 2.

The above results indicate that the relation of the amplitudes and phase lags of the reflected Kelvin wave with the incident wave plays a decisive role in the tidal system in the KS, especially in the formation of amphidromic points, for both the $K_1$ and $M_2$ tides.



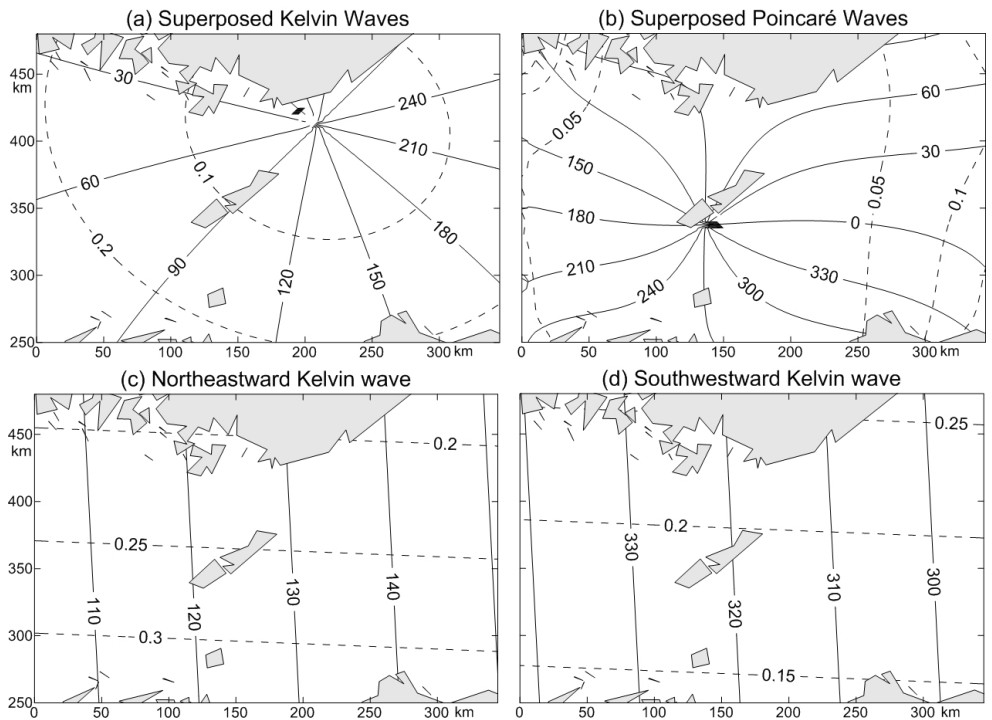

**Figure 7: Decomposed charts for the model-produced K₁ tide in the Korea Strait: (a) contribution of Kelvin waves; (b) contribution of Poincaré modes; (c) northeastward (incident) Kelvin wave; and (d) southwestward (reflected) Kelvin wave.**

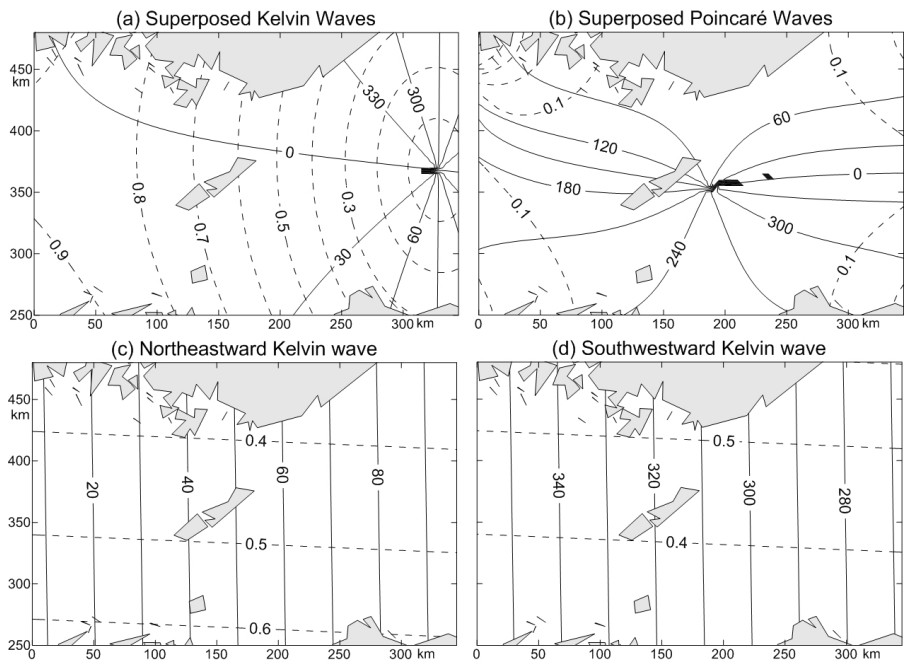

5       **Figure 8: Same as in Fig. 7 but for M₂.**



## 4 Discussion on the formation mechanism of amphidromic points

To explore the tidal dynamics of the KS–JS basin, especially the formation mechanism of amphidromic points, we consider the simplest case: a one-dimensional tidal model in channels. In the one-dimensional case, the amphidromic point is equivalent to the wave node. As previously mentioned, an important feature of the topography of the KS–JS basin is that there is a sharp

continental slope between the KS and JS, and to northeast of this slope, the JS is much deeper and wider than the KS. Thus, the channel is idealized to contain two areas, with the first one (Area1) having uniform depth $h_1$ and uniform width $W_1$ and the second one (Area2) having uniform depth $h_2$ and uniform width $W_2$. Therefore, the idealized channel contains abrupt changes in depth and width at the connection of these two areas. An incident wave enters the first area and propagates toward the second area passing over the topographic step. For simplicity, we neglect friction.

Dean and Dalrymple (1984) have presented a solution for a tidal waves travelling in such a channel; however, in their study, the wave was allowed to radiate out from the second area freely, which implies that the second area is assumed to be semi-infinitely long. Their solution shows that a part of the wave is reflected at the connecting point and another part is transmitted into the second area. The amplitude of the transmitted wave is

$$H_T = \kappa_T H_I, \tag{28}$$

where $H_I$ is the amplitude of the incident wave and $\kappa_T$ is called the transmission coefficient, which is equal to

$$\kappa_T = \frac{2}{1+\rho}, \tag{29}$$

where

$$\rho = \frac{c_2 W_2}{c_1 W_1} = \frac{\sqrt{h_2}}{\sqrt{h_1}} \frac{W_2}{W_1} \tag{30}$$

with $c_j = \sqrt{gh_j}$ representing the wave speed in the $j$-th area, $j = 1, 2$. $c_j$ is in fact proportional to $\sqrt{h_j}$. The amplitude of

the reflected wave $H_R$ is

$$H_R = \kappa_R H_I \tag{31}$$

where $\kappa_R$ is called the reflection coefficient, and is equal to the following:

$$\kappa_R = \frac{1-\rho}{1+\rho} \tag{32}$$

If $\rho > 1$, namely, if $\sqrt{h_2} W_2 > \sqrt{h_1} W_1$, then $\kappa_R < 0$, (32) can be rewritten in the form

$$\kappa_R = \frac{\rho-1}{\rho+1} \exp(-i\pi). \tag{33}$$

The above equation indicates that at the connecting point, the reflected wave changes its phase lag by 180°. Therefore, the superposition of incident and reflected waves in Area1 has the minimum amplitude at the connecting point. This theory explains how the reflected wave can be generated by abrupt increases in water depth and basin width, and why the reflected wave there has a phase lag opposite to the incident wave.





The complete solution for this case is as follows (see Appendix for derivation):

$$\begin{cases} \zeta(x) = H_I\left(\exp\{-i[k_1(x-l_1)+\theta_1]\} + \frac{\rho-1}{\rho+1}\exp\{-i[-k_1(x-l_1)+2\chi_1+\theta_1+\pi]\}\right), & l_1 \ll x \ll l_2 \\ \qquad \zeta(x) = \frac{2}{1+\rho}H_I\exp\{-i[k_2(x-l_2)+\chi_1+\theta_1]\}, & l_2 \ll x \end{cases}$$

(34)

where $\theta_1$ represents the phase lag of the incident wave at the opening of Area1; $k_j = \sigma/c_j$ is the wave number, with $c_j = \sqrt{gh_j}$ representing the wave speed in Areaj, $j$=1, 2; and $\chi_1 = k_1L_1$. This solution for the $K_1$ and $M_2$ constituents for $h_1$=99 m, $L_1$=350 km, $W_1$=230 km, $h_2$=2039 m, and $W_2$=700 km is plotted with the blue curves in Fig. 9.

However, Sect. 3.3 shows that the phase-lag changes of the reflected waves relative to the incident waves are not exactly equal to 180° but rather are smaller than 180°, and the discrepancy increases with the decreasing angular frequency. To explain this discrepancy, we improve the above theory by introducing the reflected wave in the second area. In fact, the JS is represented with a semi-closed area in the two-dimensional model (Sect. 3.1), namely, all boundaries except those connected to KS are solid ones (Fig. 4). Therefore, in the following one-dimensional model, the second area is closed at its right end so that the reflection will occur at this end. In this case, the solution becomes more complicated and is dependent on the length of the second area $L_2$. The reflection coefficient $\kappa_R$ now has the following form (see Appendix for derivation):

$$\kappa_R = \exp(-i2\delta).$$

(35)

in which $\delta$ is determined by the following equations:

$$\begin{cases} \cos\delta = \frac{1+\cos 2\chi_2}{[(1+\cos 2\chi_2)^2+(\rho\sin 2\chi_2)^2]^{1/2}}, \\ \sin\delta = \frac{\rho\sin 2\chi_2}{[(1+\cos 2\chi_2)^2+(\rho\sin 2\chi_2)^2]^{1/2}}, \end{cases}$$

(36)

where $\chi_2 = k_2L_2$.

The complete solution for this case is as follows:

$$\begin{cases} \qquad \zeta(x) = H_I(\exp\{-i[k_1(x-l_1)+\theta_1]\} + \exp\{-i[-k_1(x-l_1)+2\chi_1+\theta_1+2\delta]\}), & l_1 \ll x \ll l_2 \\ \zeta(x) = \epsilon H_I(\exp\{-i[k_2(x-l_2)+(\chi_1+\phi+\theta_1)]\} + \exp\{-i[-k_2(x-l_2)+(2\chi_2+\chi_1+\phi+\theta_1)]\}), & l_2 \ll x \ll l_3 \end{cases}$$

(37)

where $\varepsilon = 2E^{-1}$. $E$ and $\phi$ are determined by the following relations:

$$\begin{cases} E\cos\phi = (\rho+1)-(\rho-1)\cos 2\chi_2, \\ \quad E\cos\phi = (\rho-1)\sin 2\chi_2. \end{cases}$$

(38)

The first terms on the rhs (right-hand side) of the two equations in Eq. (37) represent the waves propagating in the positive $x$ direction, and the second terms are those propagating in the negative $x$ direction. This solution for the $K_1$ and $M_2$ constituents for the case $h_1$=99 m, $L_1$=350 km, $W_1$=230 km, $h_2$=2039 m, $L_2$=1150 km, and $W_2$=700 km is plotted with the red curves in Fig. 9.

Equation (35) indicates that the amplitude of the reflected wave in the first area is equal to that of the incident wave. This result is natural because friction is not considered and no dissipation is present during wave propagation. Equation (35) also indicates that the phase lag of the reflected wave at the connecting point is greater than that of the incident wave at the same





point by $2\delta$. Since the node of the superposition of the incident and reflected waves appears at the place where the phase lags of these two waves are opposite, the first node should appear at $\Delta x$ away from the connecting point with

$$\Delta x = (\pi - 2\delta)/(2k_1). \tag{39}$$

The above relationship can also be obtained from the first equation of Eq. (37). The dependence of $2\delta$ on $\sigma$ for the case

$h_1$=99 m, $L_1$=350 km, $W_1$=230 km, $h_2$=2039 m, $L_2$=1150 km, and $W_2$=700 km is plotted in Fig. 10. This figure shows that

$2\delta = 0$ when $\sigma = 0$ and $2\delta$ increases with increasing $\sigma$, although it is always less than 180°. In particular, $2\delta = 167.7°$

when $\sigma = \sigma_{K_1}$ and $2\delta = 176.2°$ when $\sigma = \sigma_{M_2}$. Based on this theory, the M$_2$ and K$_1$ amphidromic points should be located

at 7.4 and 45.9 km away from the connecting point, respectively. Compared with the two-dimensional model results given in

Sect. 3.3, this theory roughly explains one third of the changes. The remaining two third of the changes can be attributed to

the effect of Coriolis force. The solution of phase-lag changes at the cross-section in the two-dimensional rotating basin

involves interactions among three Kelvin waves (an incident and a reflected Kelvin waves in Area1 and a transmitted Kelvin

wave in Area2) and two families of Poincaré modes at the connecting cross-section (one family in each area). Taylor (1922),

Fang and Wang (1966), and Thiebaux (1988) have studied the Kelvin-wave reflection at the closed cross-section of semi-

infinite rotating two-dimensional channels. In their studies, only two Kelvin waves and one family of Poincaré modes were

involved. In comparison to their studies, the present problem is much more complicated. Because of the complexity of the

problem, we will presently leave it for a future study.

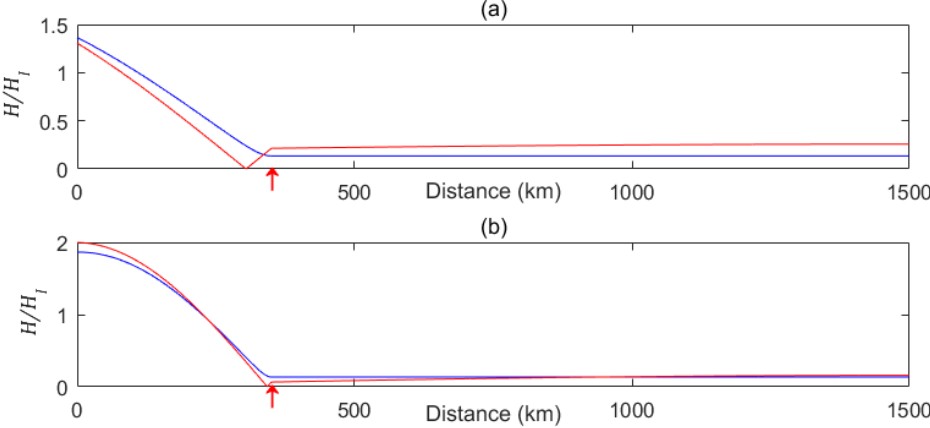

**Figure 9: Amplitude distribution along the channel. (a) K$_1$ and (b) M$_2$. Blue/red curves are solutions for semi-infinite/finite Area2.**
**The red arrow indicates the position of the connecting point between the Korea Strait and the Japan Sea. Amplitudes are given as ratios to the incident wave in Area1.**


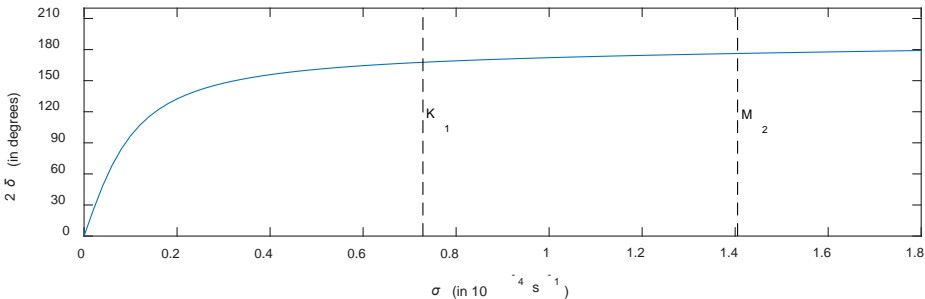

**Figure 10: Phase-lag increase of the reflected wave relative to the incident wave as a function of the angular frequency at the connecting point. See the text for details.**

## 5 Summary

In this paper, we establish a theoretical model for the KS-JS basin using the extended Taylor method. The model idealizes the study region as three connected flat rectangular areas, incorporates the effects of the Coriolis force and bottom friction in the governing equations and is forced by observed tides at the opening of the KS. The analytical solutions of the $K_1$ and $M_2$ tidal waves are obtained using Defant's collocation approach.

The theoretical model results are consistent with the satellite altimeter and tidal gauge observations, which indicates that the

model is suitable and correct. The model well reproduces the $K_1$ and $M_2$ tidal systems in the KS. In particular, the model-produced locations of the $K_1$ and $M_2$ amphidromic points are consistent with the observed ones.

The model solution provides the following insights into the tidal dynamics in the KS. (1) The tidal system in each rectangular area can be decomposed into two oppositely travelling Kelvin waves and two families of Poincaré modes, with Kelvin waves dominating the tidal system due to narrowness of the area. (2) The incident Kelvin wave from the ECS through the opening of

the KS travels toward the JS and is reflected at the connecting cross-section between the KS and JS, where abrupt increases from the KS to JS in water depth and basin width occur. (3) The phase lag of the reflected wave at the connecting cross-section increases by less than 180° relative to that of the incident wave, thus enabling the formation of the amphidromic points in the KS. (4) The phase-lag increase of the reflected wave relative to the incident wave is dependent on the angular frequency of the wave and becomes smaller as the angular frequency decreases. This feature explains why the $K_1$ amphidromic point is located

farther away from the connecting cross-section in comparison to the $M_2$ amphidromic point.

A one-dimensional model is also given in this paper to reveal the underlying basic dynamics of tides in the KS.



**Appendix: Tidal wave propagation in channels with abrupt depth/width changes.**

**a. Basic Equations**

We study tidal wave propagation in channels with abrupt depth/width changes. To be specific, we consider a one-dimensional problem corresponding to the model shown in Fig. 3. For simplicity, Area3 is combined into Area2, and the Coriolis force and

friction are neglected, then Eqs. (10) and (11) in the Sect. 2.2 of the text can be simplified as follows:

$$u_{1,-}(x) = -a_1 \exp[ik_1(x - l_1)] \tag{A1}$$

$$\zeta_{1,-}(x) = p_1 a_1 \exp[ik_1(x - l_1)] \tag{A2}$$

$$u_{1,+}(x) = b_1 \exp[-ik_1(x - l_1)] \tag{A3}$$

$$\zeta_{1,+}(x) = p_1 b_1 \exp[-ik_1(x - l_1)] \tag{A4}$$

$$u_{2,-}(x) = -a_2 \exp[ik_2(x - l_2)] \tag{A5}$$

$$\zeta_{2,-}(x) = p_2 a_2 \exp[ik_2(x - l_2)] \tag{A6}$$

$$u_{2,+}(x) = b_2 \exp[-ik_2(x - l_2)] \tag{A7}$$

$$\zeta_{2,+}(x) = p_2 b_2 \exp[-ik_2(x - l_2)] \tag{A8}$$

where $k_j = \sigma/c_j$ is the wave number, with $c_j = \sqrt{gh_j}$ representing the wave speed in Areaj, $j$=1, 2; $p_j = \sqrt{h_j/g}$; $l_1$ is the

$x$ coordinate at the opening of Area1; and $l_2 = l_1 + L_1$ is the $x$ coordinate of the connecting point of Area1 and Area2, where an abrupt change in depth and/or width occurs. In Eqs. (A1) to (A8), we have changed the notations $u_{j,1}$, $\zeta_{j,1}$, $u_{j,2}$ and $\zeta_{j,2}$ from Eqs. (10) and (11) to $u_{j,-}$, $\zeta_{j,-}$, $u_{j,+}$, and $\zeta_{j,+}$ ($j$=1, 2), respectively, to indicate the directions of wave propagation. That is, $\zeta_{j,+}(x)$ and $u_{j,+}(x)$ represent the complex amplitudes of tidal level and tidal current of the tidal waves that travel in the positive $x$ direction in Areaj, respectively; and $\zeta_{j,-}(x)$ and $u_{j,-}(x)$ represent those travelling in the negative $x$ direction

in Areaj, respectively.

The open boundary condition at $x = l_1$ can be specified as follows:

$$\zeta_{1,+}(l_1) = H_I \exp(-i\theta_1), \tag{A9}$$

where $H_I$ and $\theta_1$ represent the amplitude and phase lag of the incident wave at the opening of Area1, respectively. From Eqs. (A9) and (A4) we obtain

$$p_1 b_1 = H_I \exp(-i\theta_1), \tag{A10}$$

and

$$\zeta_{1,+}(x) = H_I \exp\{-i[k_1(x - l_1) + \theta_1]\}. \tag{A11}$$

Therefore,

$$\zeta_{1,+}(l_2) = H_I \exp[-i(\chi_1 + \theta_1)], \tag{A12}$$

where

$$\chi_1 = k_1 L_1. \tag{A13}$$





The matching conditions at $x = l_2 = l_1 + L_1$ are as follows:

$$\zeta_{1,+}(l_2) + \zeta_{1,-}(l_2) = \zeta_{2,+}(l_2) + \zeta_{2,-}(l_2), \tag{A14}$$

and

$$[u_{1,+}(l_2) + u_{1,-}(l_2)]h_1 W_1 = [u_{2,+}(l_2) + u_{2,-}(l_2)]h_2 W_2. \tag{A15}$$

To use the relationship among tidal elevations instead of tidal currents, we multiply Eq. (A15) by $p_1/h_1 W_1$ and obtain

$$\zeta_{1,+}(l_2) - \zeta_{1,-}(l_2) = \rho[\zeta_{2,+}(l_2) - \zeta_{2,-}(l_2)], \tag{A16}$$

where

$$\rho = \frac{p_1 h_2 W_2}{p_2 h_1 W_1} = \frac{\sqrt{h_2} W_2}{\sqrt{h_1} W_1} . \tag{A17}$$

**b. Solution for the case with semi-infinite Area2**

Here, we first investigate a simpler case that has been previously studied by Dean and Dalrymple (1984). In this case, Area2 is assumed to be semi-infinitely long so that the wave can propagate freely in the positive $x$ direction without reflection, meaning that $a_2 = 0$. Thus, the terms $\zeta_{2,-}$ in Eqs. (A6), (A14) and (A16) are all equal to zero. From Eqs. (A14) and (A16) with $\zeta_{2,-}(l_2) = 0$ we obtain

$$\zeta_{1,-}(l_2) = \kappa_R \zeta_{1,+}(l_2), \tag{A18}$$

and

$$\zeta_{2,+}(l_2) = \kappa_T \zeta_{1,+}(l_2), \tag{A19}$$

where $\kappa_R$ and $\kappa_T$ are called reflection and transmission coefficient respectively. These coefficients are equal to the following:

$$\kappa_R = \frac{1-\rho}{1+\rho} , \tag{A20}$$

and

$$\kappa_T = \frac{2}{1+\rho} , \tag{A21}$$

If $\rho > 1$, namely, if $\sqrt{h_2} W_2 > \sqrt{h_1} W_1$, then $\kappa_R < 0$. It is more desired to write Eq. (A20) in the following form:

$$\kappa_R = \frac{\rho-1}{\rho+1}\exp(-i\pi), \tag{A22}$$

which is Eq. (33) in the text.

   From Eqs. (A2), (A12), (A18) and (A22) we obtain

$$\zeta_{1,-}(x) = \frac{\rho-1}{\rho+1}H_I\exp\{-i[-k_1(x - l_1) + 2\chi_1 + \theta_1 + \pi]\}, \tag{A23}$$

and from Eqs. (A8), (A12), (A19) and (A21) we obtain

$$\zeta_{2,+}(x) = \frac{2}{1+\rho}H_I \exp\{-i[k_2(x - l_2) + \chi_1 + \theta_1]\}. \tag{A24}$$

Finally, we obtain the following solution:



$$\begin{cases} \zeta(x) = H_I \left( \exp\{-i[k_1(x - l_1) + \theta_1]\} + \frac{\rho-1}{\rho+1}\exp\{-i[-k_1(x - l_1) + 2\chi_1 + \theta_1 + \pi]\} \right), & l_1 \ll x \ll l_2, \\ \zeta(x) = \frac{2}{1+\rho}H_I \exp\{-i[k_2(x - l_2) + \chi_1 + \theta_1]\}, & l_2 \ll x, \end{cases} \tag{A25}$$

which is Eq. (34) in the text.

**c. Solution for the case with finite Area2**

In the following, we investigate a more complicated case that is more suitable to the KS-JS basin. In this case, Area2 is closed

at its right end, and a boundary condition is thus involved:

$$u_{2,+}(l_2 + L_2) + u_{2,-}(l_2 + L_2) = 0. \tag{A26}$$

From Eqs. (A5) to (A8), we see that Eq. (A26) is equivalent to the following:

$$\zeta_{2,+}(l_2 + L_2) - \zeta_{2,-}(l_2 + L_2) = 0. \tag{A27}$$

This further gives us

$$b_2 = a_2 \exp(i2\chi_2), \tag{A28}$$

where

$$\chi_2 = k_2 L_2. \tag{A29}$$

Hence we have

$$\zeta_{2,-}(l_2) = \zeta_{2,+}(l_2)\exp(-i2\chi_2). \tag{A30}$$

Therefore, Eqs. (A14) and (A16) can be rewritten in the following respective forms:

$$\zeta_{1,+}(l_2) + \zeta_{1,-}(l_2) = [1 + \exp(-i2\chi_2)]\zeta_{2,+}(l_2), \tag{A31}$$

and

$$\zeta_{1,+}(l_2) - \zeta_{1,-}(l_2) = \rho[1 - \exp(-i2\chi_2)]\zeta_{2,+}(l_2). \tag{A32}$$

Eliminating $\zeta_{2,+}(l_2)$ in above two equations results in

$$\zeta_{1,-}(l_2) = \frac{[1+\exp(-i2\chi_2)]-\rho[1-\exp(-i2\chi_2)]}{[1+\exp(-i2\chi_2)]+\rho[1-\exp(-i2\chi_2)]}\zeta_{1,+}(l_2). \tag{A33}$$

A few steps of algebra give us

$$\frac{1-\exp(-i2\chi_2)}{1+\exp(-i2\chi_2)} = \frac{i\sin 2\chi_2}{1+\cos 2\chi_2}. \tag{A34}$$

Substitution of Eq. (A34) in Eq. (A33) yields

$$\zeta_{1,-}(l_2) = \frac{1+\cos 2\chi_2 - i\rho\sin 2\chi_2}{1+\cos 2\chi_2 + i\rho\sin 2\chi_2}\zeta_{1,+}(l_2). \tag{A35}$$

Let

$$\begin{cases} \cos\delta = \frac{1+\cos 2\chi_2}{[(1+\cos 2\chi_2)^2+(\rho\sin 2\chi_2)^2]^{1/2}}, \\ \sin\delta = \frac{\rho\sin 2\chi_2}{[(1+\cos 2\chi_2)^2+(\rho\sin 2\chi_2)^2]^{1/2}}, \end{cases} \tag{A36}$$

then (A35) reduces to

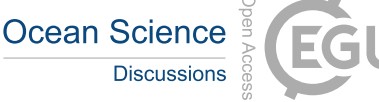

$$\zeta_{1,-}(l_2) = \zeta_{1,+}(l_2) \exp(-i2\delta), \tag{A37}$$

which is an equivalent form of Eq. (34) in the text.

From Eqs. (A4) and (A10) we have

$$\zeta_{1,+}(x) = H_I \exp\{-i[k_1(x - l_1) + \theta_1]\}, \tag{A38}$$

and from Eqs. (A12) and (A37) we have

$$\zeta_{1,-}(l_2) = H_I \exp[-i(\chi_1 + \theta_1 + 2\delta)]. \tag{A39}$$

Meanwhile, Eq. (A2) gives us

$$\zeta_{1,-}(l_2) = p_1 a_1 \exp(i\chi_1). \tag{A40}$$

Comparison of Eq. (A40) with Eq. (A39) gives

$$p_1 a_1 = H_I \exp[-i(2\chi_1 + \theta_1 + 2\delta)]. \tag{A41}$$

On substituting Eq. (41) into Eq. (A2) we have

$$\zeta_{1,-}(x) = H_I \exp\{-i[-k_1(x - l_1) + 2\chi_1 + \theta_1 + 2\delta]\}. \tag{A42}$$

From Eqs. (A31) and (A32) we obtain

$$\zeta_{2,+}(l_2) = 2[(\rho + 1) - (\rho - 1)\cos 2\chi_2 + i(\rho - 1)\sin 2\chi_2]^{-1} \zeta_{1,+}(l_2). \tag{A43}$$

Let

$$\begin{cases} E\cos\phi = (\rho + 1) - (\rho - 1)\cos 2\chi_2, \\ \quad E\sin\phi = (\rho - 1)\sin 2\chi_2, \end{cases} \tag{A44}$$

and

$$\varepsilon = 2E^{-1}, \tag{A45}$$

then (A43) reduces to

$$\zeta_{2,+}(l_2) = \varepsilon \exp(-i\phi)\zeta_{1,+}(l_2). \tag{A46}$$

Inserting Eq. (A12) into Eq. (A46) yields

$$\zeta_{2,+}(l_2) = \varepsilon H_I \exp[-i(\chi_1 + \phi + \theta_1)]. \tag{A47}$$

From Eq. (A8), we know that $p_2 b_2 = \zeta_{2,+}(l_2)$, thus we further have

$$\zeta_{2,+}(x) = \varepsilon H_I \exp\{-i[k_2(x - l_2) + (\chi_1 + \phi + \theta_1)]\}. \tag{A48}$$

Likewise, we can obtain the following solution for $\zeta_{2,-}(x)$ from Eqs. (A6) and (A47):

$$\zeta_{2,-}(x) = \varepsilon H_I \exp\{-i[-k_2(x - l_2) + (2\chi_2 + \chi_1 + \phi + \theta_1)]\}. \tag{A49}$$

Finally from Eqs. (A38), (A42), (A48) and (A49), we obtain the solution for $\zeta(x)$:

$$\begin{cases} \quad \zeta(x) = H_I(\exp\{-i[k_1(x - l_1) + \theta_1]\} + \exp\{-i[-k_1(x - l_1) + 2\chi_1 + \theta_1 + 2\delta]\}), & l_1 \ll x \ll l_2, \\ \zeta(x) = \epsilon H_I(\exp\{-i[k_2(x - l_2) + (\chi_1 + \phi + \theta_1)]\} + \exp\{-i[-k_2(x - l_2) + (2\chi_2 + \chi_1 + \phi + \theta_1)]\}), & l_2 \ll x \ll l_3, \end{cases} \tag{A50}$$

which is Eq. (37) in the text.





We have also solved the problem with the channel containing three areas corresponding to the idealized domain shown in Fig. 3. The solution is quite cumbersome and does not show significant differences from the above two-area solution (for example, it gives $2\delta = 167.8°$ when $\sigma = \sigma_{K_1}$; and $2\delta = 176.3°$ when $\sigma = \sigma_{M_2}$); therefore, the details of the solution are not given here.

*Date availability.* The ETOPO1 data (doi: 10.7289/V5C8276M) is from the National Geophysical Center, USA (https://www.ngdc.noaa.gov/mgg/global/). The DTU10 tidal data is from DTU Space, Danish National Space Center, Technical University of Denmark (ftp://ftp.space.dtu.dk/pub/DTU10).

*Author contributions.* GF conceived the study scope and the basic dynamics. DW performed calculation and prepared the draft. ZW and XC checked model results.

*Competing interests.* The authors declare that they have no conflict of interest.

*Acknowledgments.* This work was supported by the National Natural Science Foundation of China under Grant Nos. 41706031 and 41821004.

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
