# Peer review of "Study of the Tidal Dynamics of the Korea Strait Using the Extended Taylor Method"

_Ocean Science, 2020_

## Referee Comment (RC1) · David Webb (Referee) · 9 Oct 2020

Overview

This is a classic semi-analytical study of a partially enclosed tidal system. The mathematics is fairly straightforward but the authors use the results to obtain a better physical understanding for the position of the amphidromes in the strait between Korea and Japan. The paper is well laid out and easy to read and understand. I think that in principal it should be published.

Main suggestions

As I said the mathematics is fairly straightforward (maybe that is why JPO rejected the m/s), so I do not think all the details are needed in the final paper. In particular I think

that the content of the appendices may be better placed in a separate document as supplementary material (a possibility with Ocean Science).

I am also concerned that this branch of oceanic literature always ignores similar studies that have occurred in related fields of physics - in particular microwave wave guides. There used to be a complaint about the different branches of physics reinventing the wheel and to a certain extent this is true here as the Coriolis term does not necessarily introduce major changes.

For that reason I suggest that the authors, who appear to be applied mathematicians, talk to someone with a physics or microwave background about reflections from discontinuities in impedance (refractive index in the case of light). This should give a bit more insight which they could usefully add to their conclusions.

As another possibility for future work I would also suggest treating all variables as complex and investigating how the solutions at key points change with complex angular velocity - to understand how the resonant properties of the system affect the solution.

Detailed comments

1. Title

I suggest "Study of the ..."

2. Page 1, line 9

Similarly "studies of the tides ..."

3. Page 1, line 23

" ... the Yellow Sea ..."

4. Page 1, line 26

Delete 'vast'.

5. Page 1, line 27

Knives are sharp, continental slopes are steep.

6. Page 2, line 18

I disagree with 'analytical', this is a semi-analytical method, using the numerical solution of a large set of equations.

7. Page 4, line 21

This is angular velocity (radians per second) Anything with frequency refers to full cycles of something.

8. Page 5, line 8

Change to 'with momentum ... "

9. Page 16, lines 10 onwards.

This is all very standard in other areas of physics as well, so I do not think the work of Dean and Dalrymple needs to be spelt out in such detail. I suggest that you just give the results you need.

10. Page 17, line 1

You do not make clear which case you are writing about - yours or that of Dean and Dalrymple.

11. Page 18, line 9 and following

"can be attributed to ...". This is a bit of a cop out, the classic response of a committee shirking responsibility. It would read better if you were disappointed about the discrepancy but that it may be due to ... .

12. Page 19, Line 21.

I would suggest you delete this line. It is doing nothing useful.

David Webb 9 October 2020

---

## Referee Comment (RC2) · Anonymous Referee #2 · 21 Oct 2020

General comments:

This paper contains an original contribution to the co-oscillating tide in Sea of Japan (East Sea) using an extended Taylor method. Writing is considered to be reasonably good with fine piece of references. However, there is an important point authors need to make correction to enhance the quality of the paper. Specifically, extension of the three sub-region model to four sub-region model is requested. Reviewer think the extension work is not difficult but considerable time around two months might be required to make correction of the content of manuscript. For that, a major revision is recommended.

Detailed comments:

Pg.4, Lines 14-20: Authors constructed a model with three sub-regions as seen in

[Figure]

Fig. 3. However, water depth of Fig.1 and tidal chart of Fig.2 indicate the necessity of including Tartar Strait region in the analytical model. Extension of the three sub-region model to the four sub-region model is requested. On the while, review think, though not much important, representing the Japan Sea (East Sea) as the Area 2 with width W1+W3 might be sufficient rather than width W2 unless the shallow water depth along the northern coastline of Japan is considered.

Pg.7, Line 16: Authors used the Collocation approach. In fact there is another approach called Galerkin approach. Briefly comment why authors used Collocation approach. Is it mainly due to its simplicity?

Pg.8, Lines 11-12: Authors state that the influence of tide-generating force on the KS is negligible. Reviewer does not agree on this statement because the influence of direct tide generating force (DTGF) on the tide in JS can be significantly large, indirectly affecting on the tide in KS even though its direct influence on the KS is small. Reviewer think co-oscillating tide may be dominant in Japan Sea (East Sea) but DTGF has some non-negligible effects.

Pg.9, Lines 10-12: In Table 1, it is noted that water depth of area 3 is 1783m, which is comparable with that of Area 2. With the model reproduction of tide in Tartar Strait shown in Fig.2 is hardly expected.

Pg.11, Lines 11-12: Authors' statement such that the model-produced tidal systems agree fairly well with the DTU10 result is reasonably acceptable. Reviewer however notices that there are some important points authors did not comment. Close examination of Fig.5 reveals that DTU10 produces amphidromic point further north than that calculated by the analytic model and that DTU10 and analytic model produces different contour patterns in Area 2 and Area 3. Reviewer thinks that these are due to neglecting the shallow Tartar Strait region in the analytic model. Again it is addressed that Area 3 is too deep and short to include the effects of presence of the Tartar Strait. According to reviewer's modeling experience, the tides in JS (East Sea) and KS vary sensitively

with change of bottom frictional coefficient in the Tartar Strait.

Pg.12, Lines 3-5: Authors state with regard to Fig. 6 that the greatest phase lag error occurred at the northernmost corner of JS due to the existence of degenerated amphidromic point near the area. This supports the necessity of developing an extended model which takes into account the shallow Tartar Strait region.

Pg.16, Line 1: Authors discussed tidal dynamics in KS-JS basin with emphasis on the amphidromic point. However, it is hard to find any discussions related to the influence of Area 2. Reviewer think this is because no meaningful contribution by Area 2. Again, it is strongly addressed that extension of the three sub-region model to the four sub-region model is required.

―――――――――――――――――――

---

## Author Comment (AC2) · 22 Nov 2020

Response (preliminary)to the Comments from Referee 2 by Guohong Fang and Di Wu

This paper contains an original contribution to the co-oscillating tide in Sea of Japan (East Sea) using an extended Taylor method. Writing is considered to be reasonably good with fine piece of references. However, there is an important point authors need to make correction to enhance the quality of the paper. Specifically, extension of the three sub-region model to four sub-region model is requested. Reviewer think the extension work is not difficult but considerable time around two months might be required to make correction of the content of manuscript. For that, a major revision is recommended.

Reply: We sincerely thank Reviewer for his carefully reading and constructive com-

ments. We plan to extend the model domain from three sub-regions to four sub-regions in the revised manuscript.

Detailed comments:

Pg.4, Lines 14-20: Authors constructed a model with three sub-regions as seen in Fig. 3. However, water depth of Fig.1 and tidal chart of Fig.2 indicate the necessity of including Tartar Strait region in the analytical model. Extension of the three sub-region model to the four sub-region model is requested. On the while, review think, though not much important, representing the Japan Sea (East Sea) as the Area 2 with width W1+W3 might be sufficient rather than width W2 unless the shallow water depth along the northern coastline of Japan is considered.

Reply: According to this comment,we plan to extend the model domain from three sub-regions to four sub-regions in the revised manuscript.Please note that we can only artificially place Area4 northeast of Area3 rather than north of Area3due to the limitation of the Taylor method. So that the Area4 cannot overlay the actual Tartar Strait.

Pg.7, Line 16: Authors used the Collocation approach. In fact there is another approach called Galerkin approach. Briefly comment why authors used Collocation approach. Is it mainly due to its simplicity?

Reply: Yes, it is mainly due to its simplicity. In Taylor's original work, he used the Fourier method, which involves the Fourier expansions at the connecting cross-sections, and thus making the solution more complicated. To our knowledge, nobody has employed the Galerkin method in the Taylor problem, though it has been widely used in the numerical computations.

Pg.8, Lines 11-12: Authors state that the influence of tide-generating force on the KS is negligible. Reviewer does not agree on this statement because the influence of direct tide generating force (DTGF) on the tide in JS can be significantly large, indirectly affecting on the tide in KS even though its direct influence on the KS is small. Reviewer

think co-oscillating tide may be dominant in Japan Sea (East Sea) but DTGF has some non-negligible effects.

Reply: This comment correctly points out a limitation of the Taylor method. The classical and the extended Taylor methods solve the homogeneous differential equations as shown in the governing equations in our manuscript (please see alsoTaylor, 1922;Hendershott and Speranza, 1971; among others). Once the DTGF is included, the governing equations will become non-homogeneous, and basic wave forms(namely the Kelvin wave and the Poincare wave) will no longer satisfy the governing equations. This is the reason why all existing studies (please see references listed in our manuscript) do not include DTGF.

To evaluatethe influence of the DTGF on the tides in the Korea Strait, we plan to numerically compute the tides in the Korea Strait and Japan/East Sea with and without DTGF, and make comparison between two results. We will report the results in the final responses.

Pg.9, Lines 10-12: In Table 1, it is noted that water depth of area 3 is 1783m, which is comparable with that of Area 2. With the model reproduction of tide in Tartar Strait shown in Fig.2 is hardly expected.

Reply: We plan to change Table 1 to include Area4, which represents the Tartar Strait. The depth of Area4 will be much shallower than Area3.

Pg.11, Lines 11-12: Authors' statement such that the model-produced tidal systems agree fairly well with the DTU10 result is reasonably acceptable. Reviewer however notices that there are some important points authors did not comment. Close examination of Fig.5 reveals that DTU10 produces amphidromic point further north than that calculated by the analytic model and that DTU10 and analytic model produces different contour patterns in Area 2 and Area 3. Reviewer thinks that these are due to neglecting the shallow Tartar Strait region in the analytic model. Again it is addressed that Area 3 is too deep and short to include the effects of presence of the Tartar Strait. According

to reviewer's modeling experience, the tides in JS (East Sea) and KS vary sensitively with change of bottom frictional coefficient in the Tartar Strait.

Reply: We accept this comment and plan to add the fourth sub-region (Area4) to represent the Tartar Strait in the revised manuscript. The water depth of Area4 will be chosen to be equal to the mean depth of main part of the Tartar Strait. As stated previously, we can only artificially place Area4 northeast of Area3 rather than north of Area3 due to the limitation of the Taylor method. So that the Area4 cannot overlay the actual Tartar Strait.

Pg.12, Lines 3-5: Authors state with regard to Fig. 6 that the greatest phase lag error occurred at the northernmost corner of JS due to the existence of degenerated amphidromic point near the area. This supports the necessity of developing an extended model which takes into account the shallow Tartar Strait region.

Reply: We expect that an amphidromic point or a degeneratedamphidromic point will present in Area4 of the revised model.

Pg.16, Line 1: Authors discussed tidal dynamics in KS-JS basin with emphasis on the amphidromic point. However, it is hard to find any discussions related to the influence of Area 2. Reviewer think this is because no meaningful contribution by Area 2. Again, it is strongly addressed that extension of the three sub-region model to the four subregion model is required.

Reply: In the text from page 17, line 6 onward in Section 4 our focus of discussion is on the role of Area2 which representing the JS. To emphasize the importance of the JS, we will insert "Eq. (36) indicates that the length, width and depth of Area2 are also important in determining the phase-lag increase of the reflected wave relative to the incident wave in Area1" in page 17 of the revised text; and add "(5) The length, width and depth of the JS is also important in determining the phase-lag increase of the reflected Kelvin wave in the KS" to the end of Section 5 (Summary) of the revised text.

**[OSD](OSD)**

---

## Author Response (AR1)

**Author's response**

**Response to Dr. David Webb by Guohong Fang and Di Wu**

Overview

This is a classic semi-analytical study of a partially enclosed tidal system. The mathematics is fairly straightforward but the authors use the results to obtain a better physical understanding for the position of the amphidromes in the strait between Korea and Japan. The paper is well laid out and easy to read and understand. I think that in principal it should be published.

Reply: We sincerely thank Dr.Webb for his careful reading of our manuscript and constructive comments and suggestions, which are of great help in improving our study. We have addressed all these comments; our responses are given below.

Main suggestions

As I said the mathematics is fairly straightforward (maybe that is why JPO rejected the m/s), so I do not think all the details are needed in the final paper. In particular I think that the content of the appendices may be better placed in a separate document as supplementary material (a possibility with Ocean Science).

Reply: The appendix has been deleted and will be submitted separately in the form of supplementary material.

I am also concerned that this branch of oceanic literature always ignores similar studies that have occurred in related fields of physics - in particular microwave wave guides. There used to be a complaint about the different branches of physics reinventing the wheel and to a certain extent this is true here as the Coriolis term does not necessarily introduce major changes.
For that reason I suggest that the authors, who appear to be applied mathematicians, talk to someone with a physics or microwave background about reflections from discontinuities in impedance (refractive index in the case of light). This should give a bit more insight which they could usefully add to their conclusions.

Reply: The behaviour of water wave reflection in a nonrotating channel is indeed similar to the microwave reflection, or the light refraction. However, when the wave propagates in a rotating channel and the period of the wave is comparable to that of Earth's rotation the Coriolis force will have significant influence on the wave propagation and reflection. As an example, we revisit the problem of the reflection of the Kelvin wave in a semi-infinite channel first studied by Taylor (1922). Taylor' result shows that when the incident Kelvin wave is reflected at the southern shore of the North Sea, a time lag of 1.4 hr occurs due to the Coriolis force. The details are given in the appendix to this response. The main conclusion is that the Coriolis parameter has significant influence on the tidal wave reflection in a semi-infinite channel. This conclusion should also hold for the case studied in the present paper.

As another possibility for future work I would also suggest treating all variables as complex and investigating how the solutions at key points change with complex angular velocity - to understand how the resonant properties of the system affect the solution.

Reply: This is a very useful suggestion for our future work. We will try to apply the complex angular velocity to the Taylor method.

Detailed comments
1. Title
I suggest "Study of the ..."

Reply: Revised as suggested.

2. Page 1, line 9
Similarly "studies of the tides ..."

Reply: Revised as suggested.

3. Page 1, line 23
" ... the Yellow Sea ..."

Reply: Revised as suggested.

4. Page 1, line 26
Delete 'vast'.

Reply: Revised as suggested.

5. Page 1, line 27
Knives are sharp, continental slopes are steep.

Reply: The word "sharp" has been replaced with "steep" in page 1, line 27 and page 16, line 4.

6. Page 2, line 18
I disagree with 'analytical', this is a semi-analytical method, using the numerical solution of a large set of equations.

Reply: The word "analytical" has been replaced with "semi-analytical".

7. Page 4, line 21
This is angular velocity (radians per second) Anything with frequency refers to full cycles of something.

Reply: The term "angular frequency" has been replaced with "angular velocity" in page 1, line 18, page

4, line 21 and page 5, line 13. (Please note that angular frequency is a synonym of angular velocity, see Weik M.H. (2000) angular frequency. In: Computer Science and Communications Dictionary. Springer, Boston, MA. https://doi.org/10.1007/1-4020-0613-6_670 ).

8. Page 5, line 8
Change to 'with momentum ... "

Reply: The article "the" has been deleted.

9. Page 16, lines 10 onwards.
This is all very standard in other areas of physics as well, so I do not think the work of Dean and Dalrymple needs to be spelt out in such detail. I suggest that you just give the results you need.

Reply: According to this comment, we have revised this paragraph as follows: "If the second area is semi-infinitely long, allowing for the wave to radiate out from the second area freely, then a part of the wave is reflected at the connecting point and another part is transmitted into the second area. The amplitude of the transmitted wave is (see e. g. Dean and Dalrymple (1984))"

10. Page 17, line 1
You do not make clear which case you are writing about - yours or that of Dean and Dalrymple.

Reply: This equation is the same as that given by Dean and Dalrymple (1984), but in a more understandable form. In order to clarify this the words "(see also Dean and Dalrymple (1984))" have been added above this equation.

11. Page 18, line 9 and following
"can be attributed to ...". This is a bit of a cop out, the classic response of a committee shirking responsibility. It would read better if you were disappointed about the discrepancy but that it may be due to ... .

Reply: The words "can be attributed to" have been replaced with "may be duo to".

12. Page 19, Line 21.
I would suggest you delete this line. It is doing nothing useful.

Reply: This line has been deleted.

**Appendix: A Short Note on the Reflection of the Kelvin Wave in a Semi-infinite Channel: Taylor's Example Revited**

**1. Introduction**
Taylor (1922) studied the tidal system in a semi-infinite channel, with especial attention paid to the refection of the Kelvin wave at the closed end of the channel. The channel he studied is semi-infinite with a width of $W$ and a uniform depth of $h$ as shown in Fig. 1. He showed that when an incident

Kelvin wave propagates into the rotating channel, the wave would be reflected at the closed end to form a reflected Kelvin wave and an amphidromic system. Meanwhile, a series of Poincare modes would be induced in the vicinity of the closed end.

[Figure]

Fig.A1 Sketch of the semi-infinite channel.

**2. Taylor's Example**

Let

$$\begin{cases} \alpha = \dfrac{f}{c}, \\ k = \dfrac{\sqrt{\sigma^2+f^2}}{c}, \end{cases} \tag{R1}$$

where $f$ is Coriolis parameter, $\sigma$ is the angular velocity of the wave, $c$ is defined as

$$c = \frac{\pi}{W}\sqrt{gh} \tag{R2}$$

with $W$ and $h$ representing the width and depth of the channel respectively. Taylor (1922) specifically computed the relationship between the incident and reflected Kelvin waves with period equal to 12 hr (equivalent to an angular velocity of $1.4544 \times 10^{-4} \mathrm{s}^{-1}$) for the case

$$\begin{cases} \alpha = 0.7, \\ k = 0.5, \end{cases} \tag{R3}$$

which corresponds to the dimensions of the North Sea. This case was referred to as Taylor's example by Brown (1973). The estimated phase-lag increase $\theta$ of the reflected Kelvin wave versus the incident Kelvin wave at the closed end of the channel was equal to 42.10° (Taylor, 1920, p. 166). This result indicates that when the incident Kelvin wave is reflected at the southern shore of the North Sea, a time lag of 1.4 hr occurs due to the Coriolis force. The value of $\theta$ was estimated again by Brown (1973), yielding $\theta$ =42.18° (see also Thiebaux, 1988, p.369).

**3. Influence of the Coriolis parameter on the reflection of the incident Kelvin wave**

To illustrate the Influence of the Coriolis parameter on the reflection of the incident Kelvin wave, we artificially change the values of the Corisolis parameter, and apply the method discribed in our paper to the semi-infinite channel shown in Fig.1. The channel is taken 463.3 km wide (corresponding to 250 nautical miles as given by Taylor (1922)) and 63.4 m deep, then we truncate the Poincare modes up to 100 terms and calculate the values of $\theta$ for various values of $f$. The result is shown with the red curve

in Fig. 2. This figure indicates that the value of $\theta$ is zero when $f = 0$, and can be up to nearly 50° when $f = 1.4 \times 10^{-4}\text{s}^{-1}$.

For the case of Tayor's example which satisfies Eq. (3) we can obtain $f = 1.1835 \times 10^{-4}\text{s}^{-1}$ through eliminating $c$ in Eq. (1) and inserting Eq. (3). This particular value of $f$ is indicated with a vertical dashed line in Fig.2, and the corresponding value of $\theta$ is 42.16°.

Fang and Wang (1966) proposed an approximate equation for $\theta$ as follows (note that the Eq. (60) of their paper is the expression for $\theta/2$):

$$\theta = \frac{8v^3}{\pi l(l^2+v^2)\sqrt{l^2+v^2-1}\,\text{th}\frac{\pi v}{2l}},\tag{R4}$$

where

$$v = \frac{f}{\sigma},\tag{R5}$$

and

$$l = \frac{c}{\sigma}.\tag{R6}$$

The values of $\theta$ derived from (4) as function of $f$ are shown in blue curve in Fig. 2. In particular, the value of $\theta$ corresponding to $f = 1.1835 \times 10^{-4}\text{s}^{-1}$ is equal to 41.68°.

Thiebaux (1988) also proposed an approximate method for calculating $\theta$. His equation has the form

$$\theta = b_1 v' + b_3 v'^3 + O(v'^5)\tag{R7}$$

where

$$v' = \frac{\sigma}{f}.\tag{R8}$$

Thiebaux (1988) did not provide any formula for calculating $O(v'^5)$. The expressions of $b_1$ and $b_3$ are quite complicated, but their values can be calculated from his Eqs. (30) and (31) and his Table 1. The values of $\theta$ derived from (7) as function of $f$ are shown in green curve in Fig. 2. In particular, the value of $\theta$ corresponding to $f = 1.1835 \times 10^{-4}\text{s}^{-1}$ is equal to 37.19°.

[Figure]

Fig. 2 The phase-lag increase ($\theta$) of the reflected Kelvin wave versus the incident Kelvin wave at the closed end as function of the Coriolis parameter ($f$) in a semi-infinite channel, which has a width of 463.3 km and a unifom depth of 63.4 m.

**4. Conclusion**

The works of Taylor (1922), Fang and Wang (1966), Brown (1973), Thiebaux (1988) and the present study all show that the Coriolis parameter has significant influence on the tidal wave reflection in a semi-infinite channel.

**References**

Brown, P. J.: Kelvin-wave reflection in a semi-infinite canal. J. Mar. Res., 31, 1-10, 1973.

Fang, G., and Wang, J.: Tides and tidal streams in gulfs, Oceanol. Limnol. Sin., 8, 60–77, 1966. (in Chinese with English abstract).

Taylor, G. I.: Tidal oscillations in gulfs and rectangular basins. Proc. London Math. Soc., Ser. 2, 20, 148-181, https://doi.org/10.1112/plms/s2-20.1.148, 1922.

Thiebaux, M. L.: Low-frequency Kelvin wave reflection coefficient. J. Phys. Oceanogr., 18, 367-372, https://doi.org/10.1175/1520-0485(1988)018<0367:LFKWRC>2.0.CO;2, 1988.

This paper contains an original contribution to the co-oscillating tide in Sea of Japan (East Sea) using an extended Taylor method. Writing is considered to be reasonably good with fine piece of references. However, there is an important point authors need to make correction to enhance the quality of the paper. Specifically, extension of the three sub-region model to four sub-region model is requested. Reviewer think the extension work is not difficult but considerable time around two months might be required to make correction of the content of manuscript. For that, a major revision is recommended.

Reply: We sincerely thank Reviewer for his carefully reading and constructive comments. We have extended the model domain from three sub-regions to four sub-regions in the revised manuscript. Please see the following for details.

Detailed comments:

Pg.4, Lines 14-20: Authors constructed a model with three sub-regions as seen in Fig. 3. However, water depth of Fig.1 and tidal chart of Fig.2 indicate the necessity of including Tartar Strait region in the analytical model. Extension of the three sub-region model to the four sub-region model is requested. On the while, review think, though not much important, representing the Japan Sea (East Sea) as the Area 2 with width W1+W3 might be sufficient rather than width W2 unless the shallow water depth along the northern coastline of Japan is considered.

Reply: According to this comment, we have extended the model domain from three sub-regions to four sub-regions in the revised manuscript. For convenience, we call the models with three sub-regions and with four sub-regions the 3-area model and the 4-area model respectively. The 4-area model domain fitting the KS and JS is shown in Fig. R1 below. Please note that we can only artificially place Area4 northeast of Area3 rather than north of Area3 due to the limitation of the Taylor method. So that the Area4 cannot overlap the actual Tartar Strait.

[Figure]

**Fig. R1: Idealized 4-area model domain fitting the Korea Strait and Japan Sea. Copied from Figure 4 of the revised manuscript.**

[Figure]

**Fig. R2: Comparison of tidal system charts. (a) K₁ and (b) M₂ tides from the present analytical model; and (c) K₁ and (d) M₂ tides from DTU10 (Chen and Andersen, 2011). Copied from Figure 5 of the revised manuscript.**

The comparison between model results and observations is shown in Fig. R2. Correspondingly, the results in Area1 (representing the KS) of the 3-area model mentioned from page 13, line 29 to page 14, line 14 in the original manuscript are replaced with the 4-area model results in the revised manuscript. The changes in Area1 are less than 0.01 m and 2° for amplitudes and phase lags of $K_1$ respectively, and less than 0.01 m and 1° for amplitudes and phase lags of $M_2$ respectively, indicating that adding Area4 does not significantly change the tidal systems in Area1.

Pg.7, Line 16: Authors used the Collocation approach. In fact there is another approach called Galerkin approach. Briefly comment why authors used Collocation approach. Is it mainly due to its simplicity?

Reply: Yes, it is mainly due to its simplicity. In Taylor's original work, he used the Fourier method, which involved the Fourier expansions at the closed cross-sections, and thus making the solution more complicated. To our knowledge, nobody has employed the Galerkin method in the Taylor problem, though it has been widely used in the numerical computations.

Pg.8, Lines 11-12: Authors state that the influence of tide-generating force on the KS is negligible. Reviewer does not agree on this statement because the influence of direct tide generating force (DTGF) on the tide in JS can be significantly large, indirectly affecting on the tide in KS even though its direct influence on the KS is small. Reviewer think co-oscillating tide may be dominant in Japan Sea (East Sea) but DTGF has some non-negligible effects.

Reply: This comment correctly points out a limitation of the Taylor method. The classical and extended Taylor methods solve the homogeneous differential equations as shown in the governing equations in

our manuscript (please see also Taylor, 1922; Hendershott and Speranza, 1971; among others). Once the DTGF is included, the governing equations will become non-homogeneous, and the basic wave forms (namely the Kelvin wave and the Poincare wave) will no longer satisfy the governing equations. This is the reason why all existing studies (please see references listed in our manuscript) do not include DTGF.

To examine the influence of the DTGF on the tides in the Korea Strait, we have numerically computed the tides in the Korea Strait and Japan/East Sea with and without DTGF using MIKE21 model, and make comparison between these two results. As an example, Fig. R3 displays the comparison of the model-produced $M_2$ tidal systems with and without DTGF.

[Figure]

**Fig. R3: Comparison of the model-produced $M_2$ tidal system charts, (a) with DTGF, and (b) without DTGF.**

As shown in our paper title, the present study focuses on the tides in the KS. To quantitatively evaluate the influence of the DTGF on the tides in the KS, we select evenly distributed 893 points in the KS as shown in Fig. R4, and calculate the root-mean-square (RMS) vector differences between two sets of model results according the following equation:

$$\Delta = \left\{ \frac{1}{K} \sum_{k=1}^{K} \left[ \left( H_{2,k} \cos G_{2,k} - H_{1,k} \cos G_{1,k} \right)^2 + \left( H_{2,k} \sin G_{2,k} - H_{1,k} \sin G_{1,k} \right)^2 \right] \right\}^{1/2} \qquad (R1)$$

in which $k = 1, 2, \dots, K$ are indices of the points shown in Fig. R4, with $K$ representing the total number of the points (=893); $H$ and $G$ are model-produced amplitude and phase lag respectively, with subscripts 1 and 2 representing the results with and without DTGF respectively. The characteristic model-produced mean amplitude with DTGF can be calculated from the following equation:

$$\bar{H} = \left( \frac{1}{K} \sum_{k=1}^{K} H_{1,k}^2 \right)^{1/2} \qquad (R2)$$

The relative difference is represented by

$$\delta = \Delta / \bar{H} \qquad (R3)$$

The results are given in Table R1 below. From Table R1 we find that the differences between the model results in the KS with and without DTGF are not significant, indicating that the KS is dominated by co-oscillating tides.

[Figure]

**Fig. R4: Distribution of the points for comparison between the model-produced results with and without DTGF.**

**Table R1. Difference and relative difference between model results with and without direct tidal generating force (DTGF)**

|  | Δ | $\bar{H}$ | δ |
|---|---|---|---|
| M2 | 0.0092 | 0.6731 | 0.0137 |
| K1 | 0.0075 | 0.1625 | 0.0459 |

Pg.9, Lines 10-12: In Table 1, it is noted that water depth of area 3 is 1783m, which is comparable with that of Area 2. With the model reproduction of tide in Tartar Strait shown in Fig.2 is hardly expected.

Reply: We have changed Table 1 to include Area4, which represents the Tartar Strait. The depth of Area4 is taken 90 m, much shallower than Area3.

Pg.11, Lines 11-12: Authors' statement such that the model-produced tidal systems agree fairly well with the DTU10 result is reasonably acceptable. Reviewer however notices that there are some important points authors did not comment. Close examination of Fig.5 reveals that DTU10 produces amphidromic point further north than that calculated by the analytic model and that DTU10 and analytic model produces different contour patterns in Area 2 and Area 3. Reviewer thinks that these are due to neglecting the shallow Tartar Strait region in the analytic model. Again it is addressed that Area 3 is too deep and short to include the effects of presence of the Tartar Strait. According to reviewer's modeling experience, the tides in JS (East Sea) and KS vary sensitively with change of bottom frictional coefficient in the Tartar Strait.

Reply: We accept this comment and add the fourth sub-region (Area4) to represent the Tartar Strait in the revised manuscript. The water depth of Area4 is taken 90 m, which is equal to the mean depth of the main part of the Tartar Strait. After adding Area4, the agreement between model results and DTU10 data is slightly improved.

Pg.12, Lines 3-5: Authors state with regard to Fig. 6 that the greatest phase lag error occurred at the northernmost corner of JS due to the existence of degenerated amphidromic point near the area. This

supports the necessity of developing an extended model which takes into account the shallow Tartar Strait region.

Reply: The 4-area model does show a degenerated amphidromic point for $M_2$ in Area4, which is consistent with observed feature as shown in Fig. R2.

Pg.16, Line 1: Authors discussed tidal dynamics in KS-JS basin with emphasis on the amphidromic point. However, it is hard to find any discussions related to the influence of Area 2. Reviewer think this is because no meaningful contribution by Area 2. Again, it is strongly addressed that extension of the three sub-region model to the four subregion model is required.

Reply: In the text of the original manuscript from page 17, line 6 onward in Section 4 our focus of discussion is on the role of Area2 which representing the JS. To emphasize the importance of the JS, we insert "Eq. (36) indicates that the length, width and depth of Area2 are also important in determining the phase-lag increase of the reflected wave relative to the incident wave in Area1" in page 17 of the revised text; and add "(5) The length, width and depth of the JS is also important in determining the phase-lag increase of the reflected Kelvin wave in the KS" to the end of Section 5 (Summary) in the revised text.

**Page 1.**

Line 1. The title "Study on "has been changed to "Study of …".

Line 9. "studies on" has been changed to "studies of".

Line 10. "…three connected uniform-depth rectangular areas…" has been changed to "…four connected uniform-depth rectangular areas…".

Line 18. "angular frequency" has been changed to "angular velocity".

Line 23. The word "the" has been added before "Yellow Sea".

Line 26. The word "vast" has been deleted.

Line 27. "A sharp continental" has been changed to "A sheep continental".

**Page 2.**

Line 18. "analytical solutions" has been changed to "semi-analytical solutions".

**Page 3.**

Line1. Figure 1 has been replaced by a new figure with "TTS", and "TTS-Tartar Strait" has been added in the caption.

**Page 4.**

Lines 2-3. "an analytical solution" has been changed to "a theoretical solution".

Line 18. Figure 3 has been replaced by a new figure with a four-rectangle structure.

Line 21. "angular frequency" has been changed to "angular velocity".

**Page 5.**

Line 8. The word "the" before "momentum advection" has been deleted.

Line 13. "angular frequency" has been changed to "angular velocity".

**Page 6.**

Line 23. "both the idealized KS and JS" has been changed to "all rectangular areas shown in Fig.3".

**Page 8.**

Line 17. "three rectangular areas" has been changed to "four rectangular areas".

Line 18. "our area of focus" has been changed to "our focus area".

Line 19. "two rectangles" has been changed to "three rectangles".

Line 22. "angular frequencies" has been changed to "angular velocities".

**Page 9.**

Lines 1-3. Figure 4 has been replaced by a new figure with a four -rectangle structure, and "A, B, … , J" in the caption has been changed to "A, B,…, M".

Lines 5-6. "… in these three areas are 2686 km, 12189 km, and 11398 km, respectively, and those of the $M_2$ Kelvin waves are 1393 km, 6321 km, and 5911 km, respectively" has been changed to "… in these

four areas are 2686 km, 12189 km, 11398 km, and 2561 km, respectively, and those of the M$_2$ Kelvin waves are 1393 km, 6321 km, 5911 km and 1328 km, respectively."

Lines 12. The fifth column about Area4 has been inserted in Table 1.

**Page 11.**

Lines 1-2. Figure 5 has been replaced by a new figure with new solutions, and "analytical model" in the caption has been changed to "theoretical model".

Lines 4-6. Figure 6 has been replaced by a new figure with new solutions, and "A, B, … , J" in the caption has been changed to "A, B, C, D, G, H, I, J K, L, M".

Line 11. The sentence "A degenerated amphidromic point appears near the entrance of the Tartar Strait." has been inserted.

**Page 12.**

Line 4. "approximately 70° at the northernmost corner of the JS" has been changed to "approximately 64° near the entrance of the Tartar Strait".

Line 9. The RMS differences "0.014 and 0.031 m" and "7.4° and 6.4°" have been changed to "0.014 and 0.032 m" and "7.0° and 5.2°", respectively.

Line 12. Table 3 has been replaced with the new solutions.

**Page 13.**

Line18. "0.96 cm" has been changed to "0.95 cm".

**Page 13 Line 30 ~Page 14 Line 14**.

All the data have been replaced with the new solutions, which is shown as follows:

"The incident and reflected K$_1$ Kelvin waves are shown in Figs. 7c and 7d, respectively. The area-mean amplitude of the incident Kelvin wave in the KS is 0.248 m, and that of the reflected Kelvin wave is 0.190 m, which is 77% of the incident Kelvin wave. On the connecting cross-section, the section-mean amplitude of the incident Kelvin wave is 0.243 m, and the section-mean phase lag is 151.6°. The section-mean amplitude of the reflected Kelvin wave is 0.194 m, which is 80% of the incident Kelvin wave. The section-mean phase lag is 295.8°, indicating that the phase lag increases by 144.2° when the wave is reflected. The amphidromic point of the superposed Kelvin wave is 137 km away from the step and close to the northwest shore of the KS.

The incident and reflected M$_2$ Kelvin waves are shown in Figs. 8c and 8d, respectively. The area-mean amplitude of the incident Kelvin wave in the KS is 0.471 m, and that of the reflected Kelvin wave is 0.439 m, which is 93% of the incident Kelvin wave. This ratio is larger than the K$_1$ tide because the bottom friction of M$_2$ is smaller and less energy is lost in the propagation process. On the connecting cross-section, the mean amplitude of the incident Kelvin wave is 0.462 m, and the phase lag is 97.9°. The mean amplitude of the reflected Kelvin wave is 0.447 m, which is 97% of the incident Kelvin wave, and the phase lag is approximately 266.4°, with a phase-lag increase of 168.5°, which is closer to 180° as compared to the corresponding value of the K$_1$ tide. Accordingly, the M$_2$ amphidromic point of the superposed Kelvin wave shifts to approximately 21 km away from the step. A comparison between Fig. 7a and Fig. 8a shows that the amphidromic point of K$_1$ is located west of that of M$_2$. This result reproduces well the observed phenomenon as seen from Fig. 2."

**Page 15.**
Lines 1-4 Figure 7 and figure 8 have been replaced with the new solutions.

**Page 16.**
Lines 4-5. "a sharp continental slope" has been changed to "a steep continental slope".
Lines 10-14. "Dean and Dalrymple (1984) have presented a solution for a tidal waves travelling in such a channel; however, in their study, 10 the wave was allowed to radiate out from the second area freely, which implies that the second area is assumed to be semi-infinitely long. Their solution shows that a part of the wave is reflected at the connecting point and another part is transmitted into the second area." has been changed to "If the second area is semi-infinitely long, allowing for the wave radiating out from the second area freely, then a part of the wave is reflected at the connecting point and another part is transmitted into the second area. The amplitude of the transmitted wave is (see e. g. Dean and Dalrymple (1984))".

**Page 17.**
Line 1. "Appendix for derivation" has been change to "also Dean and Dalrymple (1984)".
Line 12. "Appendix" has been changed to "supplement"
Line 16 The sentence "Eq. (36) indicates that the length, width and depth of Area2 are also important in determining the phase-lag increase of the reflected wave relative to the incident wave in Area1." has been added after "$\chi_2 = k_2 L_2$."

**Page 18.**
Line 9. "can be attributed to" has been changed to "may be due to".

**Page 19**
Line 20. "(5) The length, width and depth of the JS is also important in determining the phase-lag increase of the reflected Kelvin wave in the KS." has been added at the end of the paragraph.
Line 21. The last sentence has been deleted.

**Pages 20-23**
The Appendix has been deleted in the manuscript, and will be submitted as a supplement.

**Page 24**

[revised manuscript text omitted]

---

## Author Response (AR2)

**Authors' Response (by Guohong Fang and Di Wu)**

Topic Editor Decision: Publish subject to minor revisions (review by editor) (09 Feb 2021) by Joanne Williams
Comments to the Author:
Dear authors,

Thank-you for your extensively revised manuscript. It has been re-reviewed by the original referee who requested the extension to 4 areas, and as you see they are satisfied with your changes. Their only further request is that you specify (p 17 & 19) the tendency of phase-lag changes of the Kelvin wave in KS according to length, width and depth change of JS. I suggest that the sensitivity would also be valuable, and that this is done in the form of specific examples, eg "if the bathymetry is increased by 1 m this leads to a shift in phase of x" etc. It should be quite quick to calculate.

Reply: Thank you very much for handling our manuscript and your decision. According to your comment, we have added the following statement to the text after page 17, line 3, in which a more representative phase-lag difference $\Delta = \pi - 2\delta$ has been introduced and calculated:
Comparison of Eq. (35) with Eq. (33) indicates that the phase-lag increase is now $2\delta$ instead of $\pi$. Their difference $\Delta = \pi - 2\delta$ characterises the influence of Area2 upon the phase-lag increase at the connection of two areas. To show the influence of the length, width and depth of Area2 on the value of $\Delta$, we first retain the width and depth unchanged and increase the length by 10%, it is shown that the value of $\Delta$ for $K_1$ is reduced by 15% (reduced to 10.44° from 12.27°), and that the value of $\Delta$ for $M_2$ is reduced by 37% (reduced to 2.37° from 3.78°). Next we retain the length and depth unchanged and increase the width by 10%, it is shown that the value of $\Delta$ for $K_1$ is reduced by 9% (reduced to 11.16° from 12.27°), and that the value of $\Delta$ for $M_2$ is reduced by 9% (reduced to 3.44° from 3.78°). Then we retain the length and width unchanged and increase the depth by 10%, it is shown that the value of $\Delta$ for $K_1$ increases by 1% (increases to 12.42° from 12.27°), and that the value of $\Delta$ for $M_2$ increases by 9% (increases to 4.12° from 3.78°).

Comments of Referee #2:
In Pg.19, lines 13-14 of the revised manuscript authors state that the length, width and depth of the JS is also important in determining the phase-lag increase of the reflected Kelvin wave in the KS. Similar statement appears in Pg.17, lines 2-3.
This statement is correct but not good enough. It is requested to include somewhere in the main results the tendency of phase-lage changes according to the length, width and depth changes.

Reply: Please see Reply to Topic Editor above.

**Modification list**

**Page 17.**

Line 3. "Comparison of Eq. (35) with Eq. (33) indicates that the phase-lag increase is now $2\delta$ instead of $\pi$. The difference $\Delta = \pi - 2\delta$ characterises the influence of Area2 upon the phase-lag increase at the connection of two areas. To show the influence of the length, width and depth of Area2 on the value of $\Delta$, we first retain the width and depth unchanged and increase the length by 10%, it is shown that the value of $\Delta$ for $K_1$ is reduced by 15% (reduced to 10.44° from 12.27°), and that the value of $\Delta$ for $M_2$ is reduced by 37% (reduced to 2.37° from 3.78°). Next we retain the length and depth unchanged and increase the width by 10%, it is shown that the value of $\Delta$ for $K_1$ is reduced by 9% (reduced to 11.16° from 12.27°), and that the value of $\Delta$ for $M_2$ is reduced by 9% (reduced to 3.44° from 3.78°). Then we retain the length and width unchanged and increase the depth by 10%, it is shown that the value of $\Delta$ for $K_1$ increases by 1% (increases to 12.42° from 12.27°), and that the value of $\Delta$ for $M_2$ increases by 9% (increases to 4.12° from 3.78°)." has been inserted.